# Beyond Uniform Sampling: Offline Reinforcement Learning with Imbalanced Datasets

**Zhang-Wei Hong**[1],[*] **Aviral Kumar**[2], **Sathwik Karnik**[1], **Abhishek Bhandwaldar**[3],
**Akash Srivastava**[3], **Joni Pajarinen**[4], **Romain Laroche**[6], **Abhishek Gupta**[5], **Pulkit Agrawal**[1]

## Abstract

Offline policy learning is aimed at learning decision-making policies using existing datasets of trajectories without collecting additional data. The primary motivation for using reinforcement learning (RL) instead of supervised learning techniques such as behavior cloning is to find a policy that achieves a higher average return than the trajectories constituting the dataset. However, we empirically find that when a dataset is dominated by suboptimal trajectories, state-of-the-art offline RL algorithms do not substantially improve over the average return of trajectories in the dataset. We argue this is due to an assumption made by current offline RL algorithms of staying close to the trajectories in the dataset. If the dataset primarily consists of sub-optimal trajectories, this assumption forces the policy to mimic the suboptimal actions. We overcome this issue by proposing a sampling strategy that enables the policy to only be constrained to "good data" rather than all actions in the dataset (i.e., uniform sampling). We present a realization of the sampling strategy and an algorithm that can be used as a plug-and-play module in standard offline RL algorithms. Our evaluation demonstrates significant performance gains in 72 imbalanced datasets, D4RL dataset, and across three different offline RL algorithms. Code is available at `https://github.com/Improbable-AI/dw-offline-rl`.

## 1 Introduction

Offline reinforcement learning (RL) [23, 27] aims to learn a decision-making policy that maximizes the expected return (i.e., the sum of rewards over time) using a pre-collected dataset of trajectories, making it appealing for applications where data collection is infeasible or expensive (e.g., recommendation systems [28]). Without loss of generality, it can be assumed that the dataset is generated from an *unknown* policy $\pi_{\mathcal{D}}(a|s)$, also known as the *behavior* policy [24]. The goal in offline RL is to learn a policy, $\pi_{\theta}(a|s)$ with parameters $\theta$, that exceeds the performance of the behavior policy. In offline RL, a widely recognized issue is the overestimation of $Q$-values for out-of-distribution state-action pairs, leading to suboptimal policies [8, 20, 22]. This stems from incomplete coverage of the state-action space in the dataset, causing the learning algorithm to consider absent states and actions during optimization.

Most state-of-the-art offline RL algorithms [7, 9, 20, 22, 25] mitigate the issue of OOD Q-values by constraining the distribution of actions of the learned policy $\pi_{\theta}(a|s)$, to be close to the distribution of actions in the dataset. This results in a generic objective with the following form:

$$\max_{\pi_{\theta}} J(\pi_{\theta}) - \alpha \mathbb{E}_{(s,a)\sim\mathcal{D}} \left[ \mathcal{C}(s,a) \right],$$

where $J(\pi_{\theta})$ denotes the expected return of the policy $\pi_{\theta}$, $\mathcal{D}$ denotes the dataset, $\mathcal{C}$ is a regularization term that penalizes the policy $\pi_{\theta}$ for deviating from the state-action pairs in the dataset, and $\alpha$ is the

---

[*]Correspondence: `zwhong@mit.edu`, ImprobableAI Lab, Massachusetts Institute of Technology[1], RAIL Lab, UC Berkeley[2], MIT-IBM Lab[3], Aalto University[4], University of Washington[5], and independent researcher[6].

37th Conference on Neural Information Processing Systems (NeurIPS 2023).

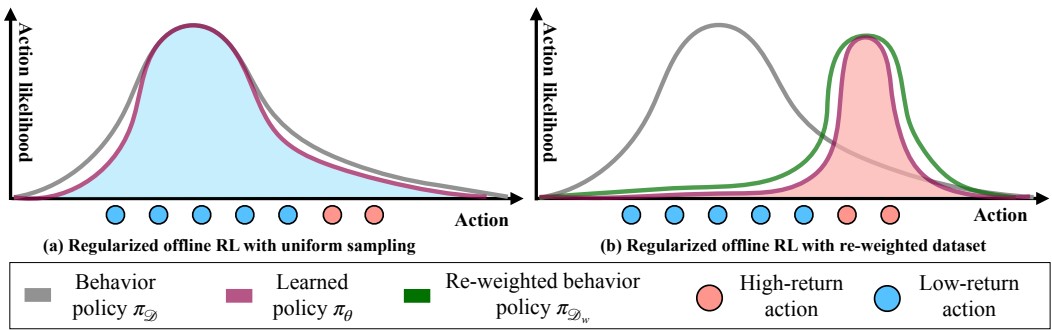

Figure 1: The dots represent actions in the dataset, where imbalanced datasets have more low-return actions. **(a)** Regularized offline RL algorithms [22, 7, 18] equally regularize the policy $\pi_\theta$ on each action, leading to imitation of low-return actions and a low-performing $\pi_\theta$. The color under the curves shows the policy's performance $J(\pi_\theta)$, with red indicating higher performance and blue indicating lower performance. **(b)** Re-weighting the dataset based on actions' returns allows the algorithm to only regularize on actions with high returns, enabling the policy $\pi_\theta$ to imitate high-return actions while ignoring low-return actions.

hyper-parameter balancing the conflicting objectives of maximizing returns while also staying close to the data distribution. This prevents offline RL algorithms from learning behaviors that produce action distributions that diverge significantly from the behavior policy.

An easy-to-understand example of choice for $\mathcal{C}$ is the squared distance between the policy and the data [7], $\mathcal{C}(s, a) := \|\pi_\theta(s) - a\|_2^2$, where $\pi_\theta(s)$ denotes the mean of the action distribution $\pi_\theta(.|s)$ and $a$ is an action sampled from the dataset. When the collected dataset is good, i.e., mostly comprising high-return trajectories, staying close to the data distribution is aligned with the objective of maximizing return, and existing offline RL algorithms work well. However, in scenarios where the dataset is skewed or imbalanced, i.e., contains only a few high-return trajectories and many low-return trajectories, staying close to the data distribution amounts to primarily imitating low-performing actions and is, therefore, detrimental. Offline RL algorithms struggle to learn high-return policies in such scenarios [12]. We present a method that overcomes this fundamental limitation of offline RL. Our method is *plug-and-play* in the sense that it is agnostic to the choice of the offline RL algorithm.

Our key insight stems from the observation that current methods are *unnecessarily conservative* by forcing the policy $\pi_\theta$ to stay close to *all* the data. Instead, we would ideally want the policy $\pi_\theta$ to be close to the *best* parts of the offline dataset. This suggests that we should constrain $\pi_\theta$ to only be close to state-action pairs that would be generated from a policy that achieves high returns, for instance, the (nearly) optimal policy $\pi^*$. In offline scenarios, where collecting additional data is prohibited, to mirror the data distribution of $\pi^*$ as much as possible, we can re-weight existing data (i.e., importance sampling [17]). We instantiate this insight in the following way: Represent the distribution induced by a better policy $\pi_{\mathcal{D}_w}$ (initially unknown) by re-weighting data points in the dataset with importance weights $w(s, a)$ and denote this distribution as $\mathcal{D}_w(s, a)$. Under this weighting, the offline RL algorithm's training objective can be written as:

$$\max_{\pi_\theta} J(\pi_\theta) - \alpha \mathbb{E}_{(s,a) \sim \mathcal{D}} \left[ w(s,a) \mathcal{C}(s,a) \right],$$

Solving for $\pi_\theta$ using the re-weighted objective constrains the policy to be close to the better policy $\pi_{\mathcal{D}}$, and, therefore, allows learning of performant policies. The key challenge is determining the weights $w$ since the state-action distribution of the better policy $\pi_{\mathcal{D}_w}$ is initially unknown. To address this, we employ off-policy evaluation techniques [32, 26, 44] to connect the data distribution $D_w$ with the expected return of the policy $\pi_{\mathcal{D}_w}$ that would generate it. This allows one to optimize the importance weights with respect to its expected return as follows:

$$\max_{w} J(\pi_{\mathcal{D}_w}) = \mathbb{E}_{(s,a) \sim \mathcal{D}_w} \left[ r(s,a) \right] \approx \mathbb{E}_{(s,a) \sim \mathcal{D}} \left[ w(s,a) r(s,a) \right].$$

Here $r(s, a)$ denotes reward of state-action pair $(s, a)$. By exploiting this connection, we optimize the importance weights $w$ to maximize the expected return $J(\pi_{\mathcal{D}_w})$ of its corresponding policy $\pi_{\mathcal{D}_w}$ subject to necessary constraints (i.e., Bellman flow conservation constraint [35, 32]). This enables us to obtain a better importance weights $w$.

We evaluate our method with state-of-the-art offline RL algorithms [22, 18] and demonstrate performance gain on 72 imbalanced datasets [12, 6]. Our method significantly outperforms prior work [12]

in challenging datasets with more diverse initial states and fewer trajectories ($20\times$ smaller than existing datasets). These datasets pose greater challenges, yet they are crucial for practical applications since real-world datasets are often small and exhibit diverse initial states (e.g., robots with different starting positions, where data comes from human teleoperation).

## 2   Preliminaries

**Typical (online) RL.** Reinforcement learning is a formalism that enables us to optimize an agent's policy in a Markov decision process (MDP [35]). The agent (i.e., decision-maker) starts from an initial state $s_0$ sampled from an initial state distribution $\rho_0(.)$. At each timestep $t$, the agent perceives the state $s_t$, takes an action $a_t \sim \pi(.|s_t)$ with its policy $\pi$, receives a reward $r_t = r(s_t, a_t)$ from the environment, and transitions to a next state $s_{t+1}$ sampled from the environment dynamics $\mathcal{T}(.|s_t, a_t)$ until reaching terminal states. The goal in RL is to learn a policy $\pi$ to maximize the $\gamma$-*discounted* expected, infinite-horizon return $J^\gamma(\pi) = \mathbb{E}_{s_0 \sim \rho_0, a_t \sim \pi(.|s_t), s_{t+1} \sim \mathcal{T}(.|s_t, a_t)} \left[ \sum_{t=0}^{\infty} \gamma^t r(s_t, a_t) \right]$. Typical (online) RL algorithms estimate the policy $\pi$'s expected return (policy evaluation) from trajectories $\tau = (s_0, a_0, r_0, s_1, a_1, r_1 \cdots)$ generated by rolling out $\pi$, and update the policy $\pi$ toward increasing $J^\gamma(\pi)$ (policy improvement), and repeat the processing by performing rollouts with the updated policy.

**Offline RL.** With no interaction with the environment allowed during the course of learning, offline RL algorithms aim to learn a policy $\pi$ that maximizes return, entirely using a fixed dataset $\mathcal{D}$ that was collected by an arbitrary and unknown "behavior policy" $\pi_\mathcal{D}$ (e.g., humans or pre-programmed controllers). These methods typically aim to estimate the return of a policy $\pi$ via techniques such as Q-learning or actor-critic, only using batches of state-action pairs $(s_t, a_t)$ uniformly drawn from $\mathcal{D}$. We will denote this estimate value of the return as $\widehat{J}^\gamma_\mathcal{D}(\pi)$. The dataset $\mathcal{D}$ consists of $N$ trajectories rolled out by $\pi_\mathcal{D}$:

$$\mathcal{D} := \{\tau_i = (s_0, a_0, r_0, s_1, a_1, r_1 \cdots s_{T_i})_i\}_{i=1}^N, \tag{1}$$

where $T_i$ denotes the length of $\tau_i$. In practice, a limit on trajectory length is required since we can only collect finite-length trajectories [34]. When the states that the policy $\pi$ would encounter and the actions that $\pi$ would take are not representative in the dataset $\mathcal{D}$, the estimated return $\widehat{J}^\gamma_\mathcal{D}(\pi)$ is typically inaccurate [8, 20, 22]. Thus, most offline RL algorithms learn the policy $\pi$ with pessimistic or conservative regularization that penalizes shift of $\pi$ from the behavior policy $\pi_\mathcal{D}$ that collected the dataset. Typically, implicitly or explicitly, the policy $\pi$ learned by most offline RL algorithms can be thought of as optimizing the following regularized objective:

$$\max_\pi \widehat{J}^\gamma_\mathcal{D}(\pi) - \alpha \mathbb{E}_{(s_t, a_t) \sim \mathcal{D}} \left[ \mathcal{C}(s_t, a_t) \right], \tag{2}$$

where $\mathcal{C}$ measures some kind of divergence (e.g., Kullback–Leibler divergence [7]) between $\pi$ and $\pi_\mathcal{D}$, and $\alpha \in \mathbb{R}^+$ denotes the strength of regularization.

## 3   Problem Statement: Unnecessary Conservativeness in Imbalanced Datasets

In this section, we describe the issue of offline RL in imbalanced datasets. While algorithms derived from the regularized offline RL objective (Equation 2) attain good performance on several standard benchmarks [6], recent work [12] showed that it leads to *"unnecessary conservativeness"* on imbalanced datasets [12] due to the use of constant regularization weight on each state-action pairs $(s, a)$ in Equation 2. To illustrate why, we start by defining *imbalance* of a dataset $\mathcal{D}$ using the positive-sided variance of the returns of the dataset (RPSV [12] defined in Definition 3.1). In essence, RPSV measures the dispersion of trajectory returns in the dataset. It indicates the room for improvement of the dataset. Figure 2 illustrates the distribution of trajectory returns in imbalanced datasets with high and low RPSV. Datasets with a low RPSV exhibit a pronounced concentration of returns around the mean value, whereas datasets with a high RPSV display a return distribution that extends away from the mean, towards higher returns. Intuitively, a dataset with high RPSV has trajectories with far higher returns than the average return of the dataset, indicating high chances of finding better data distribution through reweighting. Throughout this paper, we will use the term *imbalanced datasets* to denote datasets with high RPSV.

**Definition 3.1** (Dataset imbalance). RPSV of a dataset, $\mathbb{V}_+[G(\tau_i)]$, corresponds to the second-order moment of the positive component of the difference between trajectory return: $G(\tau_i) := \sum_{t=0}^{T_i-1} \gamma^t r(s_t^i, a_t^i)$ and its expectation, where $\tau_i$ denote trajectory in the dataset:

$$\mathbb{V}_+[G(\tau_i)] \doteq \mathbb{E}_{\tau_i \sim \mathcal{D}}\left[ (G(\tau_i) - \mathbb{E}_{\tau_i \sim \mathcal{D}}[G(\tau_i)])_+^2 \right] \quad \text{with} \quad x_+ = \max\{x, 0\}, \tag{3}$$

Imbalanced datasets are common in real-world scenarios, as collecting high-return trajectories is often more costly than collecting low-return ones. An example of an imbalanced offline dataset is autonomous driving, where most trajectories are from average drivers, with limited data from very good drivers. Due to the dominance of low-return trajectories, state-action pairs $(s, a)$ from these trajectories are over-sampled in Equation 2. Consequently, optimizing the regularized objective (Equation 2) would result in a policy that closely imitates the actions from low-performing trajectories that constitute the majority of dataset $\mathcal{D}$, but ideally, we want the policy to imitate actions only on state-action pairs from high-performing trajectories. How-

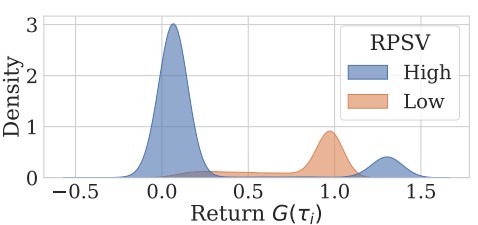

Figure 2: Return distribution of datasets with high and low RPSV. Low RPSV datasets have returns centered at the mean, while high RPSV datasets have a wider distribution extending towards higher returns. See Appendix A.4 for details.

ever, current offline RL algorithms [21, 18, 7] use constant regularization weight $\alpha$ (Equation 2). As a result, each state-action pairs is weighted equally, which leads the algorithm to be unnecessarily conservative on all data (i.e., imitating all actions of state-action pairs in the dataset). Further analysis of imbalanced datasets can be found in Appendix A.1.

## 4 Mitigating Unnecessary Conservativeness By Weighting Samples

In this section, we seek to develop an approach to address the unnecessary conservativeness issue (Section 3) of regularized offline RL algorithms in imbalanced datasets. Adding more experiences from high-performing policies to the dataset would regularize the policy to keep close to high-performing policies and hence easily mitigate the unnecessary conservativeness issue. Though collecting additional experiences (i.e., state-action pairs $(s, a)$) from the environment is prohibited in offline RL, importance sampling [32] can emulate sampling from another dataset $\mathcal{D}_w$ since the weighting can be regarded as the density ratio shown below:

$$w(s, a) = \frac{\mathcal{D}_w(s, a)}{\mathcal{D}(s, a)}, \tag{4}$$

where $\mathcal{D}_w(s, a)$ and $\mathcal{D}(s, a)$ denote the probability density of state-action pairs $(s, a)$ in dataset $\mathcal{D}_w$ and $\mathcal{D}$. Note that $\mathcal{D}_w$ is unknown but implicitly defined through a given weighting function $w$. This allows us to adjust the sampling distribution that we train the policy $\pi$ to, as suggested in the following equivalence:

$$\max_\pi \widehat{J}_{\mathcal{D}}^\gamma(\pi) - \alpha \mathbb{E}_{(s_t, a_t) \sim \mathcal{D}}\left[w(s, a)\mathcal{C}(s_t, a_t)\right] \iff \max_\pi \widehat{J}_{\mathcal{D}}^\gamma(\pi) - \alpha \mathbb{E}_{(s_t, a_t) \sim \mathcal{D}_w}\left[\mathcal{C}(s_t, a_t)\right]. \tag{5}$$

The remaining question is: *how can we determine the weighting $w(s, a)$ so that we emulate sampling from a better dataset $\mathcal{D}_w$ collected by a policy that achieves higher return than the behavior policy $\pi_{\mathcal{D}}$ that collected the original dataset $\mathcal{D}$?*.

### 4.1 Optimizing the Weightings: Emulating Sampling from High-Performing Policies

Our goal is to discover a weighting function $w$ that can emulate drawing state-action samples from a better dataset $\mathcal{D}_w$ that is collected by an *alternative behavior policy* $\pi_{\mathcal{D}_w}$ with higher return than the behavior policy $\pi_{\mathcal{D}}$ that collected the original dataset $\mathcal{D}$ (i.e., $J^\gamma(\pi_{\mathcal{D}_w}) \geq J^\gamma(\pi_{\mathcal{D}})$). We make use of density-ratio-based off-policy evaluation methods [32, 29, 44] to determine if a weighting function corresponds to a high-return policy. Note that we do not propose a new off-policy evaluation approach but rather apply the existing off-policy evaluation technique in our problem. By using these

techniques, we can relate the weighting $w$ to the expected return of the alternative behavior policy $J(\pi_{\mathcal{D}_w})$ via importance sampling formulation as follows:

$$J^\gamma(\pi_{\mathcal{D}_w}) \approx \mathbb{E}_{(s,a)\sim\mathcal{D}_w}\left[r(s,a)\right] = \mathbb{E}_{(s,a)\sim\mathcal{D}}\left[w(s,a)r(s,a)\right]. \tag{6}$$

In Equation 6, $J^\gamma(\pi_{\mathcal{D}_w})$ evaluates the quality of a given weighting function $w$. It also provides a feasible objective to optimize $w$, since it only requires obtaining samples from the original dataset $\mathcal{D}$. However, it is important to note that Equation 6 measures $\gamma$-discounted return only when the the dataset $\mathcal{D}_w$ represents a stationary state-action distribution that satisfies Bellman flow conservation constraint [35, 32, 44] in the MDP, as shown in the following equation:

$$\mathcal{D}_w(s') = (1-\gamma)\rho_0(s') + \gamma\sum_{s,a}\mathcal{T}(s'|s,a)\mathcal{D}_w(s,a) \,\forall s' \in \mathcal{S}, \quad \mathcal{D}_w(s') := \sum_{a'\in\mathcal{A}}\mathcal{D}_w(s',a') \tag{7}$$

where $\mathcal{S}$ and $\mathcal{A}$ denote the state and action spaces, respectively, and the discount factor $\gamma$ determines what discount factor corresponds to in $J^\gamma(\pi_{\mathcal{D}_w})$ in Equation 6. We slightly abuse the notation, denoting state marginal as $\mathcal{D}_w(s')$.

To estimate $J^\gamma(\pi_{\mathcal{D}_w})$ from the weighting function $w$, it is required to impose Bellman flow conservation constraint (Equaton 7) on $w$. However, it is difficult to impose this constraint due to the dependence of initial state distribution $\rho_0$ in Equaton 7. Estimating $\rho_0$ from the first state of each trajectory in the dataset is an option, but it is infeasible when the trajectories do not consistently start from initial states sampled from the distribution $\rho_0$. While we could make the assumption that all trajectories begin from initial states sampled from $\rho_0$, it would limit the applicability of our method to datasets where trajectories start from arbitrary states. We thus choose not to make this assumption since current offline RL algorithms do not require it.

Instead, since the Bellman flow conservation constraint (Equation 7) only depends on the initial state distribution $\rho_0$ when $\gamma \neq 1$, it is possible to bypass this dependence, if we maximize the undiscounted return $J(\pi_{\mathcal{D}_w}) = J^{\gamma=1}(\pi_{\mathcal{D}_w})$ (i.e., setting $\gamma = 1$ in Equation 6) of the alternative behavior policy $\pi_{\mathcal{D}_w}$. While it deviates from the RL objective presented in Equation 2, undiscounted return is often more aligned with the true objective in various RL applications, as suggested in [13]. Many RL algorithms resort to employing discounted return as an approximation of undiscounted return instead due to the risk of divergence when estimating the undiscounted return using Q-learning [39]. Thus, we constrain the weighting function $w$ to satisfy the Bellman flow conservation constraint with $\gamma = 1$ as shown below:

$$\mathcal{D}_w(s') = \sum_{s,a}\mathcal{T}(s'|s,a)\mathcal{D}_w(s,a) \,\forall s' \in \mathcal{S}. \tag{8}$$

To connect the constraint in Equation 8 to the objective in Equation 6, we rewrite Equation 8 in terms of weightings $w$ according to [29, 32][2], as shown below:

$$\mathcal{D}(s')w(s') = \sum_{s,a}\mathcal{T}(s'|s,a)w(s,a) \,\forall s' \in \mathcal{S}, \qquad w(s) := \sum_{a\in\mathcal{A}}\frac{\mathcal{D}_w(s,a)}{\mathcal{D}(s,a)} \tag{9}$$

where $w(s)$ denotes state marginal weighting. Putting the objective (Equation 6) and the constraint (Equation 9) together, we optimize $w$ to maximize the undiscounted expected return of the corresponding alternative behavior policy $\pi_{\mathcal{D}_w}$, as shown in the following:

$$\max_w J(\pi_{\mathcal{D}_w}) = \mathbb{E}_{(s,a)\sim\mathcal{D}}\left[w(s,a)r(s,a)\right] \tag{10}$$

$$\text{subject to } \mathbb{E}_{(s,a,s')\sim\mathcal{D}}\left[w(s') - w(s,a) \mid s'\right] = 0 \,\,\forall s' \in \mathcal{S}.$$

As the weightings $w$ can be viewed as the density ratio (Equation 4), we call our method as **Density-ratio Weighting (DW)**. We then re-weight offline RL algorithm, as shown in Equation 5. Note that while these weights correspond to $(s, a)$ in the dataset, this is sufficient to reweight policy optimization for offline RL.

## 4.2 Practical Implementation

**Optimizing weightings.** We begin by addressing the parameterization of the weighting function $w(s, a)$ and its state marginal $w(s')$ in Equation 10. Though state marginal $w(s')$ can derived from

---

[2]See Equation 24 in [29]

summing $w(s, a)$ over action space $\mathcal{A}$, as defined in Equation 9, it can difficult to take summation over a continuous or infinite action space. Thus we opt to parameterize the weightings $w(s, a)$ and its state marginal $w(s')$ separately. By using the identities [32] $\mathcal{D}_w(s, a) = \mathcal{D}_w(s)\pi_{\mathcal{D}_w}(a|s)$ and $\mathcal{D}(s, a) = \mathcal{D}(s)\pi_{\mathcal{D}}(a|s)$, we can represent $w(s, a)$ as the product of two ratios:

$$w(s, a) \doteq \frac{\mathcal{D}_w(s, a)}{\mathcal{D}(s, a)} = \frac{\mathcal{D}_w(s)\pi_{\mathcal{D}_w}(a|s)}{\mathcal{D}(s)\pi_{\mathcal{D}}(a|s)} = \frac{\mathcal{D}_w(s)}{\mathcal{D}(s)} \times \frac{\pi_{\mathcal{D}_w}(a|s)}{\pi_{\mathcal{D}}(a|s)}. \tag{11}$$

Michel et al. [31] showed that ratios can be parameterized by neural networks with exponential output. Thus, we represent state-action weighting $w(s, a)$ as $w_{\phi,\psi}(s, a)$ and its state marginal as $w_\phi(s)$, as shown below:

$$w_{\phi,\psi}(s, a) = \exp\phi(s)\exp\psi(s, a), \qquad\qquad w_\phi(s) = \exp\phi(s) \tag{12}$$

where $\phi$ and $\psi$ are neural networks. Next, we present how to train both neural network models. As the dataset often has limited coverage on state-action space, it is preferable to add a KL-divergence regularization $D_{KL}(\mathcal{D}_w||\mathcal{D})$ to the objective in Equation 10, as proposed in Zhan et al. [43]. This regularization keeps the state-action distribution $\mathcal{D}_w$ induced by the learned weighting $w$ close to the original dataset $\mathcal{D}$, preventing $w_{\phi,\psi}(s, a)$ from overfitting to a few rare state-action pairs in $\mathcal{D}$. Note that this does not prevent the learned weightings to provide a better data distribution for regularized offline RL algorithms. See Appendix A.2 for the detailed discussion. Another technical difficulty on training $w$ is that it is difficult to impose Bellman flow conservation constraint in Equation 10 at every state in the state space since only limited coverage of states are available in the dataset. Thus, we instead use penalty method [3] to penalize the solution of $w_{\phi,\psi}$ on violating this constraint in expectation. As a result, we optimize $w_{\phi,\psi}$ for Equation 10 using stochastic gradient ascent to optimize the following objective (details can be found in Appendix A.3):

$$\max_{\phi,\psi} \mathbb{E}_{(s,a,s')\sim\mathcal{D}}\left[\underbrace{w_{\phi,\psi}(s, a)r(s, a)}_{\text{Return}} - \lambda_F\underbrace{(w_\phi(s') - w_{\phi,\psi}(s, a))^2}_{\text{Bellman flow conservation penalty}}\right] - \lambda_K\underbrace{D_{KL}(\mathcal{D}_w||\mathcal{D})}_{\text{KL regularization}}, \tag{13}$$

where $s'$ denotes the next state observed after taking action $a$ at state $s$, and $\lambda_F, \lambda_K \in \mathbb{R}^+$ denote the strength of both penalty terms. Note that the goal of our work is not to propose a new off-policy evaluation method, but to motivate ours in the specific objective to optimize the importance weighting for training offline RL algorithms. Importantly, our approach differs from previous off-policy evaluation methods [32, 26, 44], as further discussed in the related works (Section 6).

**Applying the weighting to offline RL.** The weighing function $w_{\phi,\psi}$ could be pre-trained before training the policy, but this would introduce another hyperparameter: the number of pretraining iterations. As a consequence, we opt to train $w_{\phi,\psi}$ in parallel with the offline RL algorithm (i.e., value functions and policy). In our experiments, we perform one iteration of offline RL update pairs with one iteration of weighting function update. We also found that weighting both $\widehat{J}_{\mathcal{D}}^\gamma(\pi)$ and $\mathcal{C}(s, a)$ at each state-action pairs sampled from the dataset $\mathcal{D}$ with $w_{\phi,\psi}(s, a)$ performs better than solely weighting the regularization term $\mathcal{C}(s, a)$. For example, when weighting the training objective of implicit Q-learning (IQL) [18] (an offline RL method), the weighted objective $J_{\mathcal{D}_w}(\pi)$ is: $\mathbb{E}_{(s,a)\sim\mathcal{D}}[w_{\phi,\psi}(s, a)A(s, a)\log\pi(a|s)]$, where $A(s, a)$ denotes advantage values. Please, see Appendix A.3 for implementation details. We hypothesize that weighting both the policy optimization objective $\widehat{J}_{\mathcal{D}}^\gamma(\pi)$ and regularization $\mathcal{C}(s, a)$ in the same distribution (i.e., same importance weights) is needed to prevent policy $\pi$ increasing $\widehat{J}_{\mathcal{D}}^\gamma(\pi)$ by exploiting out-of-distribution actions on states with lower weights $w_{\psi,\phi}(s, a)$, which could lead to poor performance [8]. Appendix A.5.5 compares weighting both and only one objective. The training procedure is outlined in Algorithm 1.

## 5 Experimental Evaluation

Our experiments aim to answer whether our density-ratio weighting (DW) (Section 4) approach can improve the performance of offline RL algorithms with different types of imbalanced datasets (Section 3). Prior work on imbalanced datasets [12] focused exclusively on imbalanced datasets with trajectories originating from a similar initial state. However, in real-world scenarios, trajectories can be collected from diverse initial states. For instance, when collecting datasets for self-driving cars, it is likely that drivers initiate the recording of trajectories from drastically different initial locations. We

---

**Algorithm 1** Density-ratio weighting with generic offline RL algorithms (details in Appendix A.3)

---

1: **Input:** Dataset $\mathcal{D}$
2: Initialize policy $\pi$ and weighting function $w_{\phi,\psi}$
3: **while** not converged **do**
4:     Sample a batch $\mathcal{B}$ of tuples of states, actions, rewards, and next states $(s, a, r, s')$ from $\mathcal{D}$
5:     Update $w_{\phi,\psi}$ with batch $\mathcal{B}$ using Equation 13
6:     Update policy $\pi$ and value function with an offline RL training objective with weights $w_{\phi,\psi}(s, a)$ and $\mathcal{B}$ (e.g., [22, 7, 18])
7: **end while**

---

found that imbalanced datasets with diverse initial states exhibit a long-tailed distribution of trajectory returns, while those with similar initial states show a bimodal distribution (see Appendix A.4 for details). As diversity of initial states affects the type of imbalance, we focus our experimentation on the two types of datasets: *(i) Trajectories with similar initial states* and *(ii) Trajectories with diverse initial states*.

Following the protocol in prior offline RL benchmarking [6], we develop representative datasets of each type using the locomotion tasks from the D4RL Gym suite. Our datasets are generated by combining $1 - \sigma\%$ of trajectories from the `random-v2` dataset (low-performing) and $\sigma\%$ of trajectories from the `medium-v2` or `expert-v2` dataset (high-performing) for each locomotion environment in the D4RL benchmark. For instance, a dataset that combines $1 - \sigma\%$ of random and $\sigma\%$ of medium trajectories is denoted as `random-medium-`$\sigma\%$. We evaluate our method and the baselines on these imbalanced datasets across four $\sigma \in \{1, 5, 10, 50\}$, four environments. Both types of datasets are briefly illustrated below and detailed in Appendix A.4. Additionally, we present the results on the rest of original D4RL datasets in Appendix A.5.

**(i) Trajectories with similar initial states.** This type of datasets was proposed in [12], mixing trajectories gathered by high- and low-performing policies, as described in Section 3. Each trajectory is collected by rolling out a policy starting from similar initial states until reaching timelimit or terminal states. We consider a variant of smaller versions of these datasets that have small number of trajectories, where each dataset contains $50,000$ state-action pairs, which is 20 times smaller. These smaller datasets can test if a method overfits to small amounts of data from high-performing policies.

**(ii) Trajectories with diverse initial states.** Trajectories in this type of dataset start from a wider range of initial states and have varying lengths. One real-world example of this type of dataset is a collection of driving behaviors obtained from a fleet of self-driving cars. The dataset might encompass partial trajectories capturing diverse driving behaviors, although not every trajectory accomplishes the desired driving task of going from one specific location to the other. As not all kinds of driving behaviors occur with equal frequency, such a dataset is likely to be imbalanced, with certain driving behaviors being underrepresented

## 5.1 Evaluation Setup

**Baselines and prior methods.** We consider uniform sampling (denoted as Uniform) as the primary baseline for comparison. In addition, we compare our method with two existing approaches for improving offline RL performance on imbalanced datasets: advantage-weighting (AW), proposed in the recent work by [12] and percentage-filtering (PF) [5]. Both AW and PF sample state-action pairs with probabilities determined by the trajectory's return to which they belong. The sampling probabilities for AW and PF are given as follows:

$$\mathcal{P}_{\text{AW}}(s_t^i, a_t^i) \propto \exp((G(\tau_i) - V_0(s_0^i))/\eta) \qquad \text{(Advantage-weighting)} \qquad (14)$$

$$\mathcal{P}_{\text{PF}}(s_t^i, a_t^i) \propto \mathbb{1}\left[G(\tau_i) \geq G_{K\%}\right] \qquad \text{(Percentage-filtering),} \qquad (15)$$

where $(s_t^i, a_t^i)$ denotes the state-action pair at timestep $t$ of trajectory $\tau_i$. $G_{K\%}$ represents a threshold for selecting the top-$K\%$ of trajectories, with $K$ chosen from $\{10, 20, 50\}$ as practiced in [12]. $V(s_0^i)$ denotes the value of the initial state $s_0^i$ in trajectory $\tau_i$, and the coefficient $\eta$ represents the temperature coefficient in a Boltzmann distribution. We consider three levels of $\eta$: low (L), medium (M), and high (H) in our experiments. Further details of the hyperparameter setup can be found in Appendix A.4. For the following experiments, we implement our DW and the above baselines on the top of state-of-the-art offline RL algorithms: Conservative Q-Learning (CQL) [22], Implicit

Q-Learning (IQL) [19], and TD3BC [7]. Note that as AW can provide a better initial sampling distribution to train the weighting function in DW, we initialize training DW with AW sampling (denoted as *DW-AW*) and initialize training DW with uniform sampling (denoted as *DW-Uniform*) in the following experiments. We refer the readers to Appendix A.3 for the implementation details.

**Evaluation metrics.** Following the settings of [6], we train all algorithm for one million gradient steps with three random seeds in each dataset. We evaluate the performance of policies acquired through each method in the environment corresponding to the dataset by conducting 20 episodes every 1000 gradient step. To determine the policy's performance at a given random seed, we compute the average returns over 20 episodes during the final 10 evaluation rounds, each round separated by 1000 gradient steps. We chose to average over the performance at the last 10 rounds rather than solely at the last evaluation round because we observed that the performance of offline RL algorithms oscillates during gradient steps. The main performance metric reported is the interquartile mean (IQM) [1] of the normalized performance across multiple datasets, along with its 95% confidence interval calculated using the bootstrapping method. As suggested in [1], IQM is a robust measure of central tendency by discarding the top and bottom 25% of samples, making it less sensitive to outliers.

## 5.2   Scenario (i): Trajectories with Similar Initial States

Figure 3a shows IQM of the normalized return for thirty-two different datasets where trajectories start from similar initial states (Section 5). For all the datasets, we use the best hyperparameters for AW and PF found in [12] and the hyperparameters for DW-AW and DW-Uniform are presented in Appendix A.4. The results demonstrate that both DW-AW and DW-Uniform outperform the uniform sampling approach confirming the effectiveness of our method. Moreover, combining DW with AW enhances the performance of DW, indicating that DW can benefit from the advantages of AW. This is likely because AW can provide a good initial sampling distribution to start training the weighting function in DW. While DW-Uniform did not exceed the performance of AW in this experiment, it should be noted that our method can be applied when datasets are not curated with trajectories such as reset-free or play style datasets where data is not collected in an episodic manner. This is useful for continuing tasks, where an agent (data curator) performs a task infinitely without termination (i.e., locomotion).

**Limited size datasets.** Figure 3b presents the results on smaller versions of 8 of these datasets used in Figure 3a. Note that as we observe the higher temperature $\eta$ enables AW with CQL to perform better in this type of dataset, we additionally consider AW-XH (extra high temperature) for comparison to provide AW with as fair a comparison point as possible. Further details on the hyperparameter settings can be found in Appendix A.4. Our methods consistently achieve significantly higher returns compared to AW and PF when combined with CQL. This suggests that our methods effectively utilize scarce data in smaller datasets better than weighted sampling approaches (AW and PF) that rely on episodic trajectory-based returns rather than purely transition level optimization like DW. In the case of IQL and TD3BC, we see a clear performance improvement of our methods over uniform sampling and PF while our methods perform on par with AW. For IQL, we hypothesize that this is because IQL is less prone to overfitting on small amounts of data due to its weighted behavior cloning objective [18], which always uses the in-distribution actions. However, it is worth noting that IQL falls short in performance compared to CQL with DW-Uniform. This suggests that IQL may primarily focus on replicating behaviors from high-return trajectories instead of surpassing them, as it lacks the explicit dynamic programming used in CQL.

**Takeaway.** Since CQL with DW-AW outperforms the other two offline RL algorithms in both dataset types, our suggestion is to opt for CQL with DW-AW, especially when dealing with datasets that might exhibit an imbalance and include trajectories originating from comparable initial states.

## 5.3   Scenario (ii): Trajectories with Diverse Initial States

Figure 4 presents the results on thirty-two datasets of trajectories with diverse initial states (Section 5). We observe that uniform sampling's performance drops significantly in these datasets compared to trajectories with similar initial states, indicating that the presence of diverse initial states exacerbates the impact of imbalance. Both of our methods consistently outperform all other approaches considered in Section 5.2, including AW and PF methods. Notably, even the best-performing variant of AW

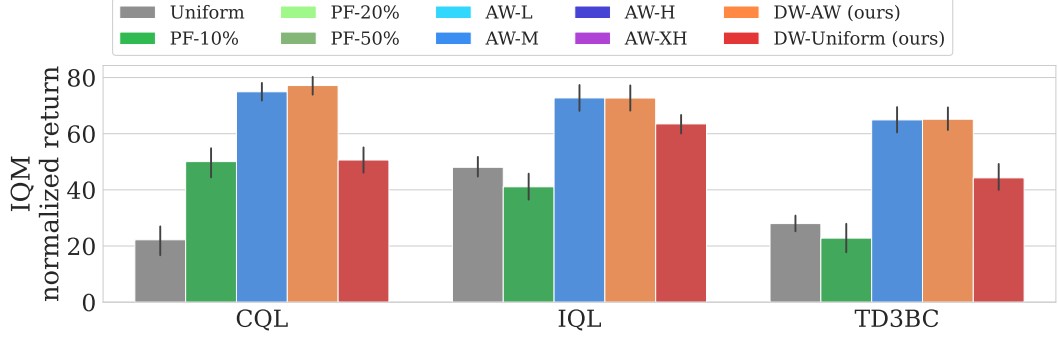

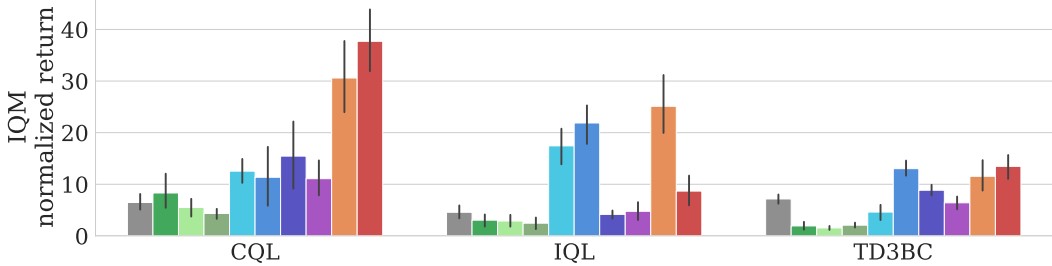

(a) Results on imbalanced datasets of trajectories with similar initial states (Section 5.1).

(b) Results on smaller version of datasets used in Figure 3a.

Figure 3: **(a)** Our methods, DW-AW and DW-Uniform, achieve higher return than Uniform, indicating that DW can enhance the performance of offline RL algorithms on imbalanced datasets. Note that our methods in IQL, although not surpassing AW and PF-10% in performance, ours can be applied to offline RL dataset that are not curated with trajectories. **(b)** Our methods outperform Uniform in CQL, IQL, and TD3BC, indicating no significant overfitting in smaller datasets. DW-AW demonstrates superior returns compared to AW and PF, particularly in CQL, indicating our method effectively leverages limited data. IQL shows limited gains likely due to its difficulties in utilizing data from the rest of low-return trajectories in the dataset (see Section 5.2).

(AW-M) falls short of matching the performance of our DW-AW, demonstrating the effectiveness of DW in leveraging the initial sampling distribution provided by AW and furthering its performance. The performance degradation of AW can be attributed to the presence of diverse initial states and varying trajectory lengths in these datasets. In such cases, state-action pairs in trajectories with high returns are not necessarily generated by high-performing policies. For instance, a sub-optimal policy can also easily reach the goal and achieve a high return if it starts close to the goal (i.e., lucky initial states). Consequently, over-sampling state-action pairs from high-return trajectories can introduce bias towards data in trajectories starting from lucky initial states. Although AW attempts to address this issue by subtracting the expected return of initial states (see Section 5.1), our results show that AW has limited success in addressing this issue. This is because the estimated expected returns of initial states can be inaccurate since AW uses the trajectories' returns in the dataset to estimate initial states' returns (i.e., Monte Carlo estimates). The trajectories in the dataset are finite in length, which makes the Monte Carlo estimates of expected returns inaccurate. To conclude, when an imbalanced dataset consists of trajectories starting from diverse initial states, we recommend using DW-AW to re-weight the training objectives in offline RL algorithms.

## 6 Related Work

Our approach builds upon recent advances in off-policy evaluation techniques, specifically density-ratio importance correction estimation (DiCE) [32]. DiCE has been primarily used for policy evaluation [29, 32, 10], while our method make use DiCE (i.e., the learned importance weights) to re-weight samples for offline RL algorithms. Recent works [43, 37, 33, 26] optimize the policy using DiCE via re-weighting behavior cloning with DiCE, while we found it fails to match offline RL algorithms' performance even in datasets with plenty of expert demonstration (Appendix A.5).

Offline imitation learning approaches [15, 30, 41] also consider imbalanced datasets similar to ours. However, these methods assume prior knowledge of which data points are generated by experts,

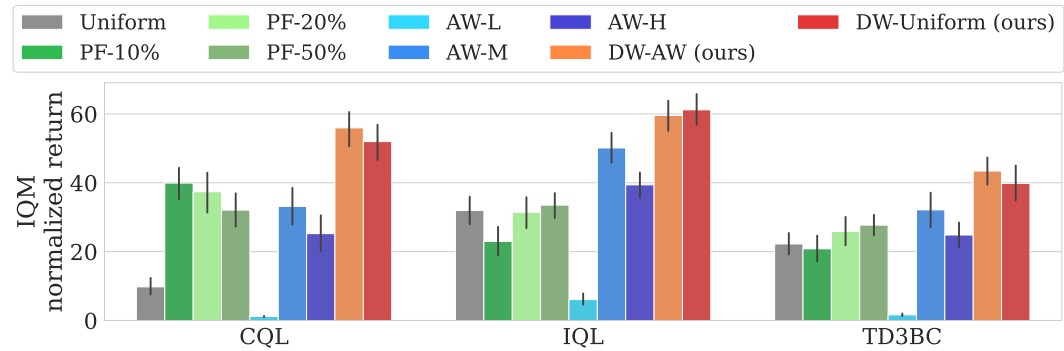

Figure 4: Results on imbalanced datasets with trajectories starting from diverse initial states (Section 5.3). Compared to Figure 3a, the performance of uniform sampling and AW decrease, showing that diverse initial states exacerbate the issue of imbalance. Our methods, DW-AW and DW-Uniform, achieve higher return than all the baselines, which suggests DW is advantageous in broader types of imbalanced datasets.

while our approach does not rely on such information. Furthermore, our method can effectively handle datasets that include a mixture of medium-level policies and low-performing policies, whereas existing approaches often rely on expert-labeled data.

Multi-task offline RL algorithms [42, 14] filter data relevant to the current task of interest from datasets collected from multiple task. For example, Yu et al. [42] employ task relevance estimation based on Q-value differences between tasks. While our motivation aligns with data filtering, our problem setting differs as we do not assume knowledge of task identifiers associated with the data points. Additionally, our dataset comprises varying levels of performance within the same task, while existing works mix data from different tasks.

Support constraints [20, 38, 2, 40] have been proposed as an alternative approach to prevent offline RL algorithms from exploiting out-of-distribution actions, distinct from distributional constraints used in state-of-the-art methods [22, 7, 18]. While support constraints theoretically suit imbalanced data, the prior work [38] found that support constraints have not shown significant improvements beyond distributional constraint-based algorithms. Note that our method is independent of the constraint used in offline RL algorithms. Thus support constraints is orthogonal to our approach.

## 7 Conclusion, Future Directions, and Limitations

Our method, density-ratio weighting (DW) improves the performance of state-of-the-art offline RL algorithms [22, 18] over 72 imbalanced datasets with varying difficulties. In particular, our method exhibits substantial improvements in more challenging and practical datasets where the trajectories in the dataset start from diverse initial states and only limited amount of data are available. Future works can explore other optimization techniques to better address the Bellman flow conservation constraint in importance weights optimization (e.g., Augmented Lagrangian method [37]). Additionally, it would be valuable to study the impact of violating this constraint on the effectiveness of importance-weighted offline RL algorithms.

**Limitations.** Although our method improves performance by optimizing sample weights, we lack theoretical guarantees due to the absence of a unified theoretical analysis on the dependence of state-of-the-art offline RL algorithms on imbalanced data distribution. While some theoretical works [4, 36] have analyzed the interplay between data distribution and offline RL algorithm performance, they primarily focus on specific algorithms that differ significantly from the practical state-of-the-art offline RL algorithms.

## Author Contributions

- **Zhang-Wei Hong:** Led the project and the writing of the paper, implemented the method, and conducted the experiments.
- **Aviral Kumar:** Advised the project in terms of theory, algorithm development, and experiment design. Revised the paper and positioned the paper in the field.
- **Sathwik Karnik:** Prepared the datasets and proofread the paper.
- **Abhishek Bhandwaldar:** Helpd scaling up experiments in the cluster.
- **Akash Srivastava:** Advised the project in the details of the practical and theoretical algorithm design and coordinated the compute.
- **Joni Pajarinen:** Advised the project in the details of the practical and theoretical algorithm design and experiment designs.
- **Romain Laroche:** Advised the project in the theory of the algorithms and dataset designs.
- **Abhishek Gupta:** Advised the project in terms of theory, algorithm development, and experiment design. Revised the paper and positioned the paper in the field.
- **Pulkit Agrawal:** Coordinated the project, revised the paper, and positioned the paper in the field.

## Acknowledgements

We thank members of the Improbable AI Lab for helpful discussions and feedback. We are grateful to MIT Supercloud and the Lincoln Laboratory Supercomputing Center for providing HPC resources. This research was supported in part by the MIT-IBM Watson AI Lab, an AWS MLRA research grant, Google cloud credits provided as part of Google-MIT support, DARPA Machine Common Sense Program, ARO MURI under Grant Number W911NF-21-1-0328, ONR MURI under Grant Number N00014-22-1-2740, and by the United States Air Force Artificial Intelligence Accelerator under Cooperative Agreement Number FA8750-19-2-1000. The views and conclusions contained in this document are those of the authors and should not be interpreted as representing the official policies, either expressed or implied, of the Army Research Office or the United States Air Force or the U.S. Government. The U.S. Government is authorized to reproduce and distribute reprints for Government purposes notwithstanding any copyright notation herein.

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

# A  Appendix

## A.1   Additional Discussion on Imbalanced Datasets

We consider imbalanced datasets that mixes the trajectories logged by high- and low- performing behavior policies, where trajectories from low-performing behavior policies predominate the dataset. When dealing with such imbalanced datasets, the regularized offline RL objective (Equation 2) tends to constrain the learned policy $\pi$ to stay close to the overall mediocre behavior policy rather than the high-performing one, which is an unnecessary form of conservativeness. We illustrate this issue from intuitive and analytical aspects in the following.

Intuitively, for example, when considering a dataset $\mathcal{D}$ that mix the trajectories from two behavior policies $\pi_{\mathcal{D}}^H$ and $\pi_{\mathcal{D}}^L$ where $\pi_{\mathcal{D}}^H$ significantly outperforms $\pi_{\mathcal{D}}^L$ (i.e., $J(\pi_{\mathcal{D}}^H) \geq J(\pi_{\mathcal{D}}^L)$), if the dataset is imbalanced such that state-action pairs $(s_t, a_t)$ logged by $\pi_{\mathcal{D}}^L$ predominate, the regularized offline RL objective will penalize deviation from $\pi_{\mathcal{D}}^L$ more than deviation from $\pi_{\mathcal{D}}^H$. This is because uniform sampling on the imbalanced dataset oversamples state-action pairs $(s_t, a_t)$ from $\pi_{\mathcal{D}}^L$, leading to an over-weighting of the regularization term on those state-action pairs $(s_t, a_t)$.

Analytically, the issue of unnecessary conservativeness (Section 3) can be explained by how the performance of offline RL $J(\pi)$ is dependent on the performance of the behavior policy $J(\pi_{\mathcal{D}})$, as suggested in [12]. In the case of an imbalanced dataset where the behavior policy can be regarded a mixture of $\pi_{\mathcal{D}}^H$ and $\pi_{\mathcal{D}}^L$, the performance of the behavior policy $J(\pi_{\mathcal{D}})$ can be estimated by the expected return of the distribution of trajectory in the dataset $\mathcal{D}$, as shown below:

$$J(\pi_{\mathcal{D}}) \approx \mathbb{E}_{\tau_i \sim \mathcal{D}} [G(\tau_i)] = \sum_{i=1}^{N} \mathcal{D}(\tau_i) G(\tau_i), \tag{16}$$

where $\mathcal{D}(\tau_i)$ denotes the probability mass of trajectory $\tau_i$ in $\mathcal{D}$, $G(\tau_i) := \sum_{t=0}^{T_i-1} r(s_t^i, a_t^i)$ is the return (i.e., sum of rewards) of trajectory $\tau_i$, and $T_i$ is its length. Since the trajectories from $\pi_{\mathcal{D}}^L$ dominate the dataset, the average return is mostly determined by low-return trajectories, leading to low $J(\pi_{\mathcal{D}})$. This prevents pessimistic and conservative algorithms [22, 7, 18, 20, 25] from achieving high $J(\pi)$ with low $J(\pi_{\mathcal{D}})$. Both types of algorithms aim to learn a policy $\pi$ that outperforms $\pi_{\mathcal{D}}$ (i.e., $J(\pi) \geq J(\pi_{\mathcal{D}})$), while constraining the policy $\pi$ to stay close to $\pi_{\mathcal{D}}$ [4]. Recent theoretical analysis by Singh et al. [38] on conservative Q-learning (CQL) [22] (an offline RL algorithm) shows that the performance improvements of the learned policy $\pi$ over the behavior policy is upper-bounded by the deviation of $\pi$ from $\pi_{\mathcal{D}}$ as shown in: $J(\pi) - J(\pi_{\mathcal{D}}) \leq C\mathbb{E}_{(s,a)\sim\mathcal{D}}[\mathcal{C}(s,a)]$, where $C$ is a constant and $\mathcal{C}$ denotes the regularization penalty (see Section 2). This implies that regularized objective would constrain the policy $\pi$ from deviating from $\pi_{\mathcal{D}}$, thus corroborating the findings in [12] despite the absence general theoretical guarantees for all offline RL algorithms.

**Why our method improves performance on imbalanced datasets?**   Our importance reweighting method can be seen as a modification of the performance of the behavior policy, which represents the lower bound in conservative and pessimistic offline RL algorithms. By adjusting the weights of state-action pairs sampled from the dataset, we simulate the sampling of data from an alternative dataset $\mathcal{D}_w$ that would be generated by an alternative behavior policy $\pi_{\mathcal{D}_w}$. By maximizing the expected return $J(\pi_{\mathcal{D}_w})$ of this alternative behavior policy $\pi_{\mathcal{D}_w}$, we can obtain a $\pi_{\mathcal{D}_w}$ that outperforms the original behavior policy $\pi_{\mathcal{D}}$, i.e., $J(\pi_{\mathcal{D}_w}) \geq J(\pi_{\mathcal{D}})$. As mentioned earlier, most offline RL algorithms aim to achieve a policy that performs at least as well as the behavior policy that collected the dataset. By training an offline RL algorithm using a dataset $\mathcal{D}_w$ induced by reweighting with $w$, we obtain a higher lower bound on the policy's performance, as $J(\pi_{\mathcal{D}_w}) \geq J(\pi_{\mathcal{D}})$.

## A.2   Why regularizing the importance weightings with KL divergence does not prevent policy improvement?

In Section 4.2, we employ a KL divergence penalty denoted as $D_{KL}(\mathcal{D}||\mathcal{D}_w)$ to impose a penalty on the extent to which the weighted data distribution $\mathcal{D}_w$ diverges from the original data distribution $\mathcal{D}$. It is important to note that this penalty does not inhibit the learned weightings from potentially improving the data distribution for regularized offline RL algorithms. Essentially, our approach involves optimizing the data distribution for offline RL algorithms first. Subsequently, these offline RL algorithms are subjected to regularization based on this optimized distribution, thereby further

enhancing the policy. This framework enables offline RL algorithms to deviate more significantly from the original data distribution.

To illustrate this concept, consider a scenario where our density-ratio weightings are constrained within an $\epsilon$ KL divergence margin with respect to the original data distribution. A regularized offline RL algorithm, which is also subject to an $\epsilon$ KL divergence constraint with respect to this weighted data distribution, can potentially improve its performance even when it deviates by more than $2\epsilon$ from the original data distribution. It is worth noting that some readers might think this as being analogous to reducing the regularization weight $\alpha$. However, reducing $\alpha$ (Equation 2) lets offline RL algorithms to determine how far they should deviate from the original data distribution on their own. In our experimental investigations, we have observed that reducing $\alpha$ often results in worse performance compared to utilizing our DW method, as evidenced by the results presented in Figure [small kl].

### A.3 Implementation details

#### A.3.1 Offline RL algorithms

We implement our method and other baselines on top of two offline RL algorithms: Conservative Q-Learning (CQL) [22] and Implicit Q-Learning (IQL) [18]. Our baselines, advantage-filtering (AW) and percentage-filtering (PF), are weighted-sampling methods (see Section 5.1). Therefore, they do not require changing the implementation of offline RL algorithms. In the following sections, we explain the details of our importance weighting method for each algorithm.

**CQL.** We reweight both the actor (policy) and the critic (Q-function) in CQL as follows:

$$\max_{\pi} \mathbb{E}_{s\sim\mathcal{D},a\sim\pi(.|s)} \left[ w_\phi(s) \left( Q(s,a) - \log\pi(a|s) \right) \right] \qquad \text{(Actor)} \qquad (17)$$

$$\min_{Q} \alpha\mathbb{E}_{(s,a)\sim\mathcal{D}} \left[ w_{\phi,\psi}(s,a) \left( \log\sum_{a'\in\mathcal{A}} \exp Q(s,a') - Q(s,a) \right) \right] + \qquad \text{(Critic)} \qquad (18)$$

$$\mathbb{E}_{(s,a,s')\sim\mathcal{D},a'\sim\pi(.|s')} \left[ w_{\phi,\psi}(s,a) \left( r(s,a) + \gamma Q(s',a') - Q(s,a) \right)^2 \right]$$

$$\min_{\alpha} \mathbb{E}_{s\sim\mathcal{D},a\sim\pi(.|s)} \left[ w_\phi(s) \left( -\log\pi(a|s) - \bar{\mathcal{H}} \right) \right] \qquad \text{(Entropy coefficient [11])}, \qquad (19)$$

where $\bar{\mathcal{H}}$ denotes the target entropy used in soft actor critic (SAC) [11] and $w_\phi$ and $w_{\phi,\psi}$ denote the weights predicted by our method (see Section 4). We follow the notations in CQL paper [22]. The implementation details of computing Equation 17 can be found in [22]. Our implementation is adpated from open-sourced implementation: `JaxCQL`[3].

**IQL.** We reweight state-value function ($V$), state-action value function (i.e,. Q-function, $Q$), and policy ($\pi$) in IQL as follows:

$$\min_{V} \mathbb{E}_{(s,a)\sim\mathcal{D}} \left[ w_\phi(s) L_2^\tau(Q(s,a) - V(s)) \right] \qquad (20)$$

$$\min_{Q} \mathbb{E}_{(s,a,s')\sim\mathcal{D}} \left[ w_{\phi,\psi}(s,a) \left( r(s,a) + \gamma V(s') - V(s) \right)^2 \right] \qquad (21)$$

$$\max_{\pi} \mathbb{E}_{(s,a)\sim\mathcal{D}} \left[ w_{\phi,\psi}(s,a) \exp\left( \beta \left( Q(s,a) - V(s) \right) \right) \log\pi(a|s) \right], \qquad (22)$$

where $L_2^\tau$ denotes the upper expectile loss [18] and $\beta$ denotes the temperature parameter for IQL. Our implementation is adapted from the official implementation[4] for implicit Q-learning (IQL) [18].

We attach our implementation in the supplementary material, where CQL and IQL implementations are in `JaxCQL` and `implicit_q_learning`, respectively.

---

[3]https://github.com/young-geng/JaxCQL
[4]https://github.com/ikostrikov/implicit_q_learning

### A.3.2 Density-weighting function

The following is the training objective of the importance weighting function $w_{\phi,\psi}$ and $w_\phi$ in our method, as described in Equation 10 (Section 4.2):

$$\max_{\phi,\psi} \mathbb{E}_{(s,a,s')\sim\mathcal{D}} \left[ \underbrace{w_{\phi,\psi}(s,a)r(s,a)}_{\text{Return}} - \lambda_F \underbrace{(w_\phi(s') - w_{\phi,\psi}(s,a))^2}_{\text{Bellman flow conservation penalty}} \right] - \lambda_K \underbrace{D_{KL}(\mathcal{D}_w||\mathcal{D})}_{\text{KL regularization}}.$$

The KL regularization term can be expressed as follows:

$$\begin{aligned}
D_{KL}(\mathcal{D}_w||\mathcal{D}) &= \sum_{s,a} \mathcal{D}_w(s,a) \log \frac{\mathcal{D}_w(s,a)}{\mathcal{D}(s,a)} \qquad\qquad (23) \\
&= \sum_{s,a} \mathcal{D}(s,a) \frac{\mathcal{D}_w(s,a)}{\mathcal{D}(s,a)} \log \frac{\mathcal{D}_w(s,a)}{\mathcal{D}(s,a)} \\
&= \mathbb{E}_{(s,a)\sim\mathcal{D}} \left[ \frac{\mathcal{D}_w(s,a)}{\mathcal{D}(s,a)} \log \frac{\mathcal{D}_w(s,a)}{\mathcal{D}(s,a)} \right] \\
&= \mathbb{E}_{(s,a)\sim\mathcal{D}} \left[ w(s,a) \log w(s,a) \right].
\end{aligned}$$

Thus, we can rewrite the training objective (Equation 10) to the follows:

$$\max_{\phi,\psi} \mathbb{E}_{(s,a,s')\sim\mathcal{D}} \left[ \underbrace{w_{\phi,\psi}(s,a)r(s,a)}_{\text{Return}} - \lambda_F \underbrace{(w_\phi(s') - w_{\phi,\psi}(s,a))^2}_{\text{Bellman flow conservation penalty}} - \underbrace{\lambda_K w_{\phi,\psi}(s,a) \log w_{\phi,\psi}(s,a)}_{\text{KL regularization}} \right].$$

$$(24)$$

The objective in Equation 24 can be optimized by minimizing the loss function $L(\phi,\psi)$ using stochastic gradient descent shown below:

$$L(\phi,\psi) := L_R(\phi,\psi) + \lambda_F L_F(\phi,\psi) + \lambda_K L_K(\phi,\psi) \qquad\qquad (25)$$

$$L_R(\phi,\psi) := -\sum_{i=1}^{B} \bar{w}_{\phi,\psi}(s_i,a_i) r(s_i,a_i) \qquad\qquad (26)$$

$$L_F(\phi,\psi) := \frac{1}{B} \sum_{i=1}^{B} \left( w_{\phi,\psi}(s_i') - w_{\phi,\psi}(s_i,a_i) \right)^2. \qquad\qquad (27)$$

$$L_K(\phi,\psi) := \sum_{i=1}^{B} \bar{w}_{\phi,\psi}(s,a) \log \bar{w}_{\phi,\psi}(s,a), \qquad\qquad (28)$$

where $B$ denote batch size and $(s_i,a_i)$ denotes the $i^{\text{th}}$ state-action pair in the batch. $\bar{w}_{\phi,\psi}(s_i,a_i)$ denotes the batch normalized weights predictions [31] that is defined as follows:

$$\bar{w}_{\phi,\psi}(s_i,a_i) := \frac{w_{\phi,\psi}(s_i,a_i)}{\sum_{j=1}^{B} w_{\phi,\psi}(s_j,a_j)}, \qquad\qquad (29)$$

where $w_{\phi,\psi}(s_i,a_i)$ denotes the importance weights predictions from the network $\phi$ and $\psi$ (see Section 4.2). As [31] suggests, applying batch-level normalization on the importance weights predictions can make the normalization requirement of importance weights, i.e., $\mathbb{E}[w] = 1$, more likely being satisfied during optimization than other approaches (e.g., adding normalization penalty term) (see [31] for details).

### A.4 Evaluation detail

### A.4.1 Training details

For both the baselines and our method, we train CQL and IQL for a total of one million gradient steps. The hyperparameters used for CQL and IQL are set to the optimal values recommended in

their publicly available implementations. As for the training of our density weighting functions $\phi, \psi$, we employ the same network architecture as the Q-value function architectures in CQL and IQL, which consists of a two-layer Multilayer Perceptron (MLP) with 256 neurons and ReLU activation in each layer. To minimize the objective defined in Equation 25, we train $\phi$ and $\psi$ using the Adam optimizer [16] with a learning rate of 0.0001 and a batch size of 256. In each gradient step of CQL and IQL, we train $\phi$ and $\psi$ for one gradient step as well.

We conducted hyperparameter search in the datasets with diverse trajectories. For AW with CQL we searched temperatue $\eta$: 0.01 (L), 0.1 (M, the best from the original paper), 1.0 (H), 5.0 (XH). For AW with IQL, we searched temperature $\eta$: 0.01 (L), 0.2 (M) (the best from the original paper), 1.0 (H), 5.0 (XH). For PF in CQL and IQL, we searched $K$ over 0.1, 0.2, and 0.5. The hyperparameter search results are all already presented in Figures 3a. For DW-AW, we use the temperature $\eta$ used in AW-M for CQL and IQL, except for small datasets. In small datasets (Figure 3b), we use the temperature used in AW-XH for DW-AW. For DW-AW and DW-Uniform, we searched $\lambda_K \in \{0.2, 1.0\}$ and $\lambda_F \in \{0.1, 1.0, 5.0\}$. We use the best found hyperparameter by the time we started the large scale experiments: $(\lambda_K, \lambda_F) = (0.2, 0.1)$ for CQL and $(\lambda_K, \lambda_F) = (1.0, 1.0)$ for IQL. We present the hyperparameter search results of DW-Uniform in Figure 2.

### A.4.2 Evaluation details

We follow the evaluation protocol used in most offline RL research [22, 7, 18]. Each offline RL algorithm and sampling method (including our DW approach) combination is trained for one million gradient steps with three random seeds in each dataset. We evaluate the policy learned by each method in the environment corresponding to the dataset for 20 episodes every 1000 gradient step. The main performance metric reported is the interquartile mean (IQM) [1] of the normalized mean return over the last 10 rounds of evaluation across multiple datasets, along with its 95% confidence interval calculated using the bootstrapping method. As suggested in [1], IQM is a robust measure of central tendency by discarding the top and bottom 25% of samples, making it less sensitive to outliers.

**Compute.** We ran all the experiments using workstations with two RTX 3090 GPUs, AMD Ryzen Threadripper PRO 3995WX 64-Cores CPU, and 256GB RAM.

### A.4.3 Dataset curation.

Following the protocol in prior offline RL benchmarking [6, 12], we develop representative datasets for Scenarios (i) and (ii) using the locomotion tasks from the D4RL Gym suite.

**Scenario (i): Datasets with trajectories starting from similar initial states.** This type of datasets was proposed in [12], mixing trajectories gathered by high- and low-performing policies, as described in Section 3. Each trajectory is collected by rolling out a policy starting from similar initial states until reaching timelimit or terminal states. As suggested in [12], these datasets are generated by combining $1 - \sigma\%$ of trajectories from the `random-v2` dataset (low-performing) and $\sigma\%$ of trajectories from the `medium-v2` or `expert-v2` dataset (high-performing) for each locomotion environment in the D4RL benchmark. For instance, a dataset that combines $1 - \sigma\%$ of random and $\sigma\%$ of medium trajectories is denoted as `random-medium-`$\sigma\%$. We evaluate our method and the baselines on these imbalanced datasets across four $\sigma \in \{1, 5, 10, 50\}$, four environments. We construct 32 of this type of datasets in all the combinations of $\{$`ant, halfcheetah, hopper, walker2d`$\} \times \{$`random-medium, random-expert`$\} \times \{\sigma \in \{1, 5, 10, 50\}\}$. Also, we consider a variant of smaller versions of these datasets that have small number of trajectories, where each dataset contains $50,000$ state-action pairs, which is 20 times smaller. These smaller datasets can test if a method overfits to small amounts of data from high-performing policies. As every method fails to suprpass random policy when $\sigma = 1$ and $\sigma = 5$, we only evaluate all the methods in $\sigma = 10$. Note that the results in $\sigma = 50$ is not different from the results in larger version of mixed dataset with $\sigma = 50$ since the amount of high-performing trajectories is still sufficient when $\sigma = 50$.

**Scenario (ii): Datasets with trajectories starting from diverse initial states.** Trajectories in this type of dataset start from a wider range of initial states and have varying lengths. This characteristic is designed to simulate scenarios where trajectories from different parts of a task are available in the dataset. One real-world example of this type of dataset is a collection of driving behaviors

obtained from a fleet of self-driving cars. The dataset might encompass partial trajectories capturing diverse driving behaviors, such as merging onto a highway, changing lanes, navigating through intersections, and parking, although not every trajectory accomplishes the desired driving task of going from one specific location to the other. As not all kinds of driving behaviors occur with equal frequency, such a dataset is likely to be imbalanced, with certain driving behaviors being underrepresented. We curate this type of datasets by adapting the datasets from Scenario (i). These datasets are created by combining $1 - \sigma\%$ of trajectory segments from the random-v2 dataset (representing low-performing policies) and $\sigma\%$ of trajectory segments from either the medium-v2 or expert-v2 dataset (representing high-performing policies) for each locomotion environment in the D4RL benchmark. Each trajectory segment is a subsequence of a trajectory in the dataset of Scenario (i). Specifically, for each trajectory $\tau_i$, a trajectory segment is selected, ranging in length from 10 to 50 timesteps within $\tau_i$. We chose the minimum and maximum lengths to be 10 and 50 timesteps, respectively, based on our observation that most locomotion behaviors exhibit intervals lasting between 10 and 50 timesteps. Since locomotion behaviors are periodic in nature, these datasets can simulate locomotion recordings from various initial conditions, such as different poses and velocities.

The datasets in Scenario (ii) capture a different form of imbalance that is overlooked in the datasets of Scenario (i). The imbalanced datasets in Scenario (i) combine full trajectories from low- and high-performing policies, starting from similar initial states. These datasets capture the imbalance resulting from the policies themselves, while overlooking the imbalance caused by different initial conditions. Even when using the same behavior policy, the initial conditions, i.e., the initial states of trajectories, can lead to an imbalance in the dataset, as certain initial conditions tend to yield higher returns compared to others. For example, an agent starting near the goal and another starting far away from the goal may achieve significantly different returns, even when following the same policy. This type of imbalance can exist in real-world datasets if certain initial conditions are oversampled during the dataset collection process.

Figure 5 presents the distribution of normalized returns for both types of datasets in both scenarios. The trajectory returns are normalized to a range of 0 to 1 using max-min normalization, specifically $(G(\tau_i) - G_{\min})/(G_{\max} - G_{\min})$, where $G_{\min}$ and $G_{\max}$ denote the minimum and maximum trajectory returns in the dataset, respectively. From Figure 5, we observe that the two types of datasets exhibit different return distributions. The return distribution in Scenario (i) (i.e., mixed) is closer to a bimodal distribution, where a trajectory is either from a high- or low-performing policy. On the other hand, the return distribution in Scenario (ii) (i.e., mixed (diverse)) follows a heavy-tailed distribution, where high-return trajectories are located in the long tails. This indicates that in addition to the imbalance resulting from the combination of trajectories from low- and high-performing policies, the presence of diverse initial states introduces another type of imbalance in the dataset, where initial states that easily lead to high returns are much less prevalent than others.

### A.5 Additional results

We present the full results in total of 113 datasets in Tables 4 and 5, including original D4RL datasets [6], mixed datasets (Figure 3a, denoted as Mixed), mixed datasets with diverse initial states (Figure 4, denoted as Mixed (diverse)), and small datasets (Figure 3b, denoted as Mixed (small)). As [12] suggested, the original D4RL datasets are not imbalanced and thus weighted sampling and importance weighting methods perform on par with uniform sampling.

#### A.5.1 Four room example for trajectory stitching

To assess DW's trajectory stitching ability, we conducted an experiment in a didactic four-room environment [26].

**Experiment setup:** See Figure 6 for the environment illustration. The agent starts from an orange initial state, traverses non-red cells, gaining +1 reward at the green goal, else zero. To test trajectory stitching, a suboptimal dataset with 1000 trajectories was generated, where none of each is optimal trajectory. Due to absence of optimal trajectories, up-weighting trajectories (i.e., AW and PF) with high-returns won't produce a state-action distribution matching the optimal policy. Thus, if a method can generate state-action distribution matching the one of the optimal policy, it indicates that the method is able to stitch trajectories because it can identify optimal state-action pairs leading to the goal even though those optimal state-action pairs are not observed in the same trajectory in the dataset.

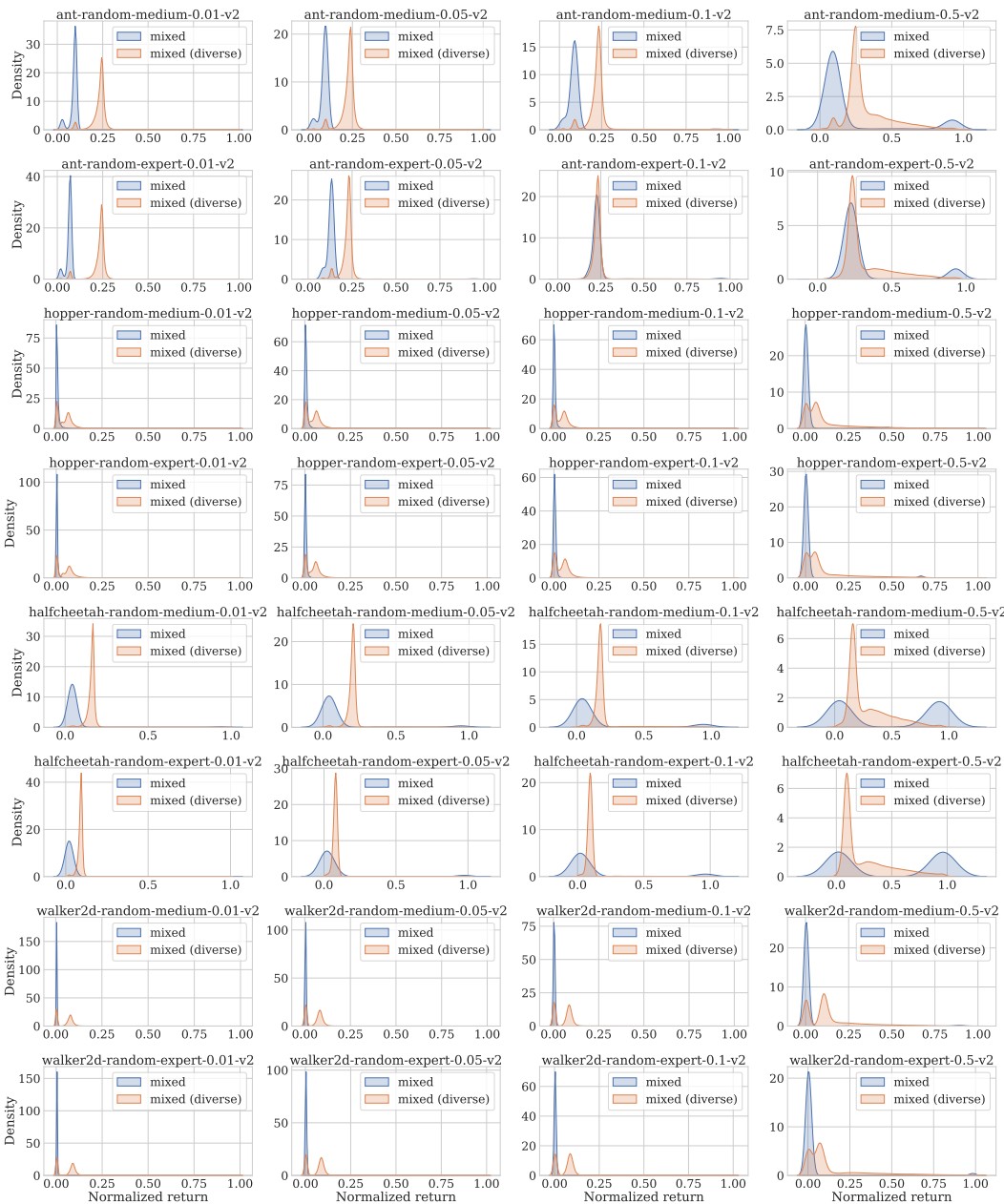

Figure 5: Return distributions of mixed datasets with similar initial states (blue, Scenario (i)) and diverse initial states (orange, Scenario (ii)). Both types of datasets lead to different kinds of imbalance and return distributions. See Section A.4.3 for details.

**Results:** Figures 6a, 6b, and 6e display the state-action distributions of behavior policy, optimal policy, and DW; the number above each plot is expected return under the state-action distribution. We see that both AW (Figure 6c) and PF (Figure 6d fail to match the state-action distribution of the optimal policy in Figure 8(b), hence leading to suboptimal performance. In contrast, DW successfully approximates the optimal policy's state-action distribution, confirming DW can identify optimal state-action pairs observed in different suboptimal trajectories and stitch them to optimal trajectories.

### A.5.2 Comparison with OptDICE

OptDiCE [26] and AlgaeDiCE [33], as well as our method, all involve learning importance weights for policy optimization. However, the usage and learning of importance weights vary across these

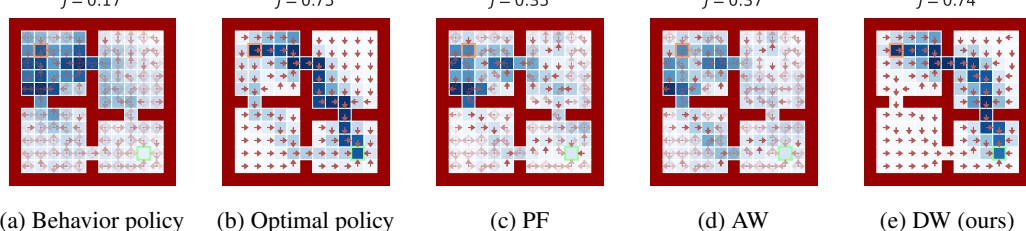

| $J = 0.17$ | $J = 0.75$ | $J = 0.35$ | $J = 0.37$ | $J = 0.74$ |
|---|---|---|---|---|
| (a) Behavior policy | (b) Optimal policy | (c) PF | (d) AW | (e) DW (ours) |

Figure 6: The stationary state-action distributions of behavior policy (a), the optimal policy (b), percentage filtering (i.e., top K%) (c), advantage weighting (d), and density weighting (e). $J$ denotes the expected return for each. The agent starts from the orange state and will receive the rewards at the green state (goal) and episode termination. Figure 6a is the empirical state-action distribution formed by 1000 suboptimal trajectories without optimal trajectory to the goal. Reweighting trajectories (i.e., AW and PF) cannot produce the optimal state-action distribution (Figure 6b) in the absence of optimal trajectories in the dataset. In contrast, since DW assign weights to transitions rather than trajectories, it can up-weight good transitions in suboptimal trajectories and stitch them to optimal state-action distribution. As we see, only DW (Figure 6e) successfully matches the optimal policy. This indicates that DW has the ability to stitch trajectories from suboptimal data distribution like Figure 6a while the other methods cannot.

methods. We compare our method with OptDiCE as it is the state-of-the-art method on policy optimization using DiCE. In the following, we illustrate the difference between our method and OptDiCE.

**Learning importance weights.** OptDiCE learns the importance weights through solving the optimization problem same as Equation 10 in an approach different from ours. OptDiCE learns the importance weights through optimizing the following primal-dual objective:

$$\min_{\nu} \max_{w \geq 0} \mathbb{E}_{(s,a)\sim\mathcal{D}}\left[e_\nu(s,a)w(s,a) - \alpha f\big(w(s,a)\big)\right] + (1-\gamma)\mathbb{E}_{s_0\sim p_0}[\nu(s_0)] \approx \tag{30}$$

$$\min_{\nu} \mathbb{E}_{(s,a,s')\sim\mathcal{D}}\left[\hat{e}_\nu(s,a,s')(f')^{-1}\big(\tfrac{1}{\alpha}\hat{e}_\nu(s,a,s')\big)_+ - \alpha f\big((f')^{-1}(\tfrac{1}{\alpha}\hat{e}_\nu(s,a,s'))_+\big)\right] + \tag{31}$$

$$(1-\gamma)\mathbb{E}_{s_0\sim p_0}[\nu(s_0)],$$

where $f$ can be any convex function (e.g., $f(x) := x\log x$), $\hat{e}_\nu(s,a,s') := r(s,a) + \gamma\nu(s') - \nu(s)$, and $x_+ := \max(0,x)$. The coefficient $\alpha$ denotes the strength of regularization: the higher the $\alpha$ is, the uniform the importance weights are. The learned importance weights $w(s,a)$ are expressed in terms of $\nu$ as follows:

$$w(s,a) = \max\left(0, (f)^{-1}\left(\frac{\hat{e}_\nu(s,a,s')}{\alpha}\right)\right). \tag{32}$$

On the other hand, our method learns the importance weights without solving the min-max optimization.

**Applying importance weights.** OptDiCE extracts the policy from the learned importance weights using information projection (I-projection) [26]. The implementation of I-projection is non-trivial, while its idea is close to weighted behavior cloning objective shown as follows:

$$\max_{\pi} \mathbb{E}_{(s,a)\sim\mathcal{D}_w}\left[\log \pi(a|s)\right] = \max_{\pi}\mathbb{E}_{(s,a)\sim\mathcal{D}}\left[w(s,a)\log\pi(a|s)\right]. \tag{33}$$

Differing from OptDiCE, we apply the importance weights to actor-critic offline RL algoritmhs like CQL [22] and IQL [18].

We present the performance of our methods in CQL and IQL, and OptDiCE in imbalanced datasets used in Scenario (i) in Figure 7, showing that OptDiCE underpeform all the other methods on imbalanced datasets and even underperforms uniform sampling approach. This result indicates that even though OptDiCE learns the importance weights by solving a similar objective with our method, OptDiCE is not effective on imbalanced datasets.

### A.5.3 Comparison with weights learned by OptDiCE

While the policy learned using OptDiCE may not be effective on imbalanced datasets, it remains uncertain whether the importance weights learned with OptDiCE can enhance the performance of

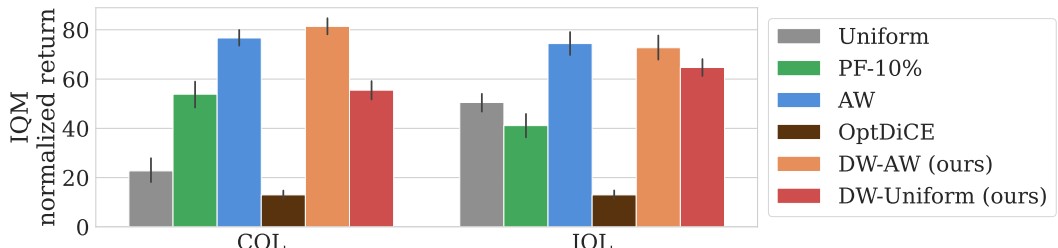

Figure 7: Results on imbalanced datasets of trajectories with similar initial states (Scenario (i) in Section 5.1). OptDiCE [26] fails to achieve high return on imbalanced datasets and even performs worse than CQL and IQL with uniform sampling. Note that OptDiCE is a density-ratio importance correction estimation (DiCE) based offline RL algorithm rather than a sampling or importance weighting method. While both bars of OptDiCE in this figure denote the same result, we plot the performance of OptDiCE alongside CQL and IQL with different sampling (or weighting) methods for comparison.

| | OptDiCEW ($\alpha = 0.1$) | OptDiCEW ($\alpha = 1.0$) | OptDiCEW ($\alpha = 5.0$) |
|---|---|---|---|
| hopper-random-expert-diverse-5%-v2 | 3.3 (-17.5) | 39.1 (+18.3) | 4.0 (-16.8) |
| hopper-random-expert-diverse-10%-v2 | 1.5 (-56.4) | 29.7 (-28.2) | 11.9 (-46.0) |
| hopper-random-medium-diverse-5%-v2 | -* | 6.4 (-58.5) | 57.3 (-7.6) |
| hopper-random-medium-diverse-10%-v2 | 2.2 (-52.9) | 30.1 (-25.0) | 41.0 (-14.1) |

(*: training gets terminated due to NaN values)

Table 1: Comparison of the weights learned by our method and that learned by OptDiCE approach. Each cell in the table denotes the mean return obtained by CQL trained with OptDiCE weights and the number in the parenthesis indicates the improvements (i.e., score$_\text{OptDiCE}$ − score$_\text{Ours}$) over the performance of DW-Uniform reported in Figure 4. It shows that OptDiCE underperforms our method in most datasets (i.e,. negative improvements), and is sensitive to the regularization strength parameter $\alpha$ (see Section A.5.3).

offline RL algorithms. To investigate this hypothesis, we reweight the training objective of CQL (Equation 17) using the importance weights learned with OptDiCE (Equation 32). We present a comparison with our method in Table 1, where OptDiCEW refers to CQL trained with OptDiCE weights, and $\alpha$ represents the regularization strength (Equation 30). We examine different values of $\alpha$ ranging from 0.1 to 5.0, but none of them consistently outperforms our method. It is worth noting that each configuration results in a significant performance loss in certain datasets. While this observation suggests that the importance weights learned using OptDiCE are not effective in improving the performance of offline RL algorithms, it is important to note that our work does not aim to propose a new off-policy evaluation method based on DiCE. Therefore, a comprehensive comparison between different methods for learning importance weights is left for future research endeavors.

### A.5.4 Hyperparameter studies

In this section, we investigate the hyperparameters of KL regularization strength ($\lambda_K$) and Bellman flow conservation strength ($\lambda_F$). We present the average returns obtained in four specific imbalanced datasets in Table 2. These datasets are chosen because they exhibit significant performance differences between uniform sampling and other methods. Our observations indicate that higher penalties for flow conservation and lower strengths of KL regularization tend to improve performance in both CQL and IQL.

### A.5.5 Comparison of re-weighting both objectives and only regularization objective

In theory, the regularization term $\mathbb{E}_{(s,a)\sim\mathcal{D}}\left[C(s,a)\right]$ (Equation 2) is the only component that can potentially harm performance on imbalanced datasets, as it encourages the policy to imitate the suboptimal actions that dominate the dataset. However, our findings suggest that reweighting all the training objectives in offline RL algorithms leads to improved performance. In this section, we

| $(\lambda_K, \lambda_F)$ | $(0.2, 0.1)$ | $(0.2, 1.0)$ | $(0.2, 5.0)$ | $(1.0, 0.1)$ | $(1.0, 1.0)$ | $(1.0, 5.0)$ |
|---|---|---|---|---|---|---|
| hopper-random-expert-diverse-5%-v2 | 20.4 | 13.8 | 22.1 | 9.6 | 7.5 | 7.9 |
| hopper-random-expert-diverse-10%-v2 | 51.7 | 27.8 | 42.1 | 10.9 | 9.8 | 5.3 |
| hopper-random-medium-diverse-5%-v2 | 64.7 | 66.1 | 60.3 | 16.2 | 5.5 | 25.4 |
| hopper-random-medium-diverse-10%-v2 | 55.4 | 58.9 | 65.3 | 3.6 | 7.2 | 7.3 |

(a) CQL

| $(\lambda_K, \lambda_F)$ | $(0.2, 0.1)$ | $(0.2, 1.0)$ | $(0.2, 5.0)$ | $(1.0, 0.1)$ | $(1.0, 1.0)$ | $(1.0, 5.0)$ |
|---|---|---|---|---|---|---|
| hopper-random-expert-diverse-5%-v2 | 14.7 | 56.4 | 54.2 | 16.8 | 68.4 | 75.1 |
| hopper-random-expert-diverse-10%-v2 | 16.2 | 67.8 | 51.0 | 78.5 | 63.0 | 67.5 |
| hopper-random-medium-diverse-5%-v2 | 53.8 | 50.8 | 51.5 | 50.9 | 49.5 | 52.6 |
| hopper-random-medium-diverse-10%-v2 | 43.4 | 48.8 | 53.0 | 42.5 | 53.2 | 50.8 |

(b) IQL

Table 2: Hyperparameter studies of our DW method in **(a)** CQL and **(b)** IQL. $\lambda_K$ and $\lambda_F$ denote the strength of KL-regularization and Bellman flow conservation penalty, respectively (see Section A.3). Lower KL-regularization strength $\lambda_F$ and higher flow conservation penalty strength $\lambda_F$ lead to better performance.

compare the performance of CQL trained with reweighted regularization only (referred to as "Reg. only") and reweighting all objectives (referred to as "All"). For the "All" approach, we train CQL using Equation 17. On the other hand, for the "Reg. only" approach, we remove the term $w_{\phi,\psi}(s,a)$ from the $\alpha$ training and the term $w_\phi(s)$ from the $\pi$ training in Equation 17. Table 3 presents the mean return of both approaches, indicating that "All" exhibits slightly better performance compared to "Reg. only".

| | Reg. only | All |
|---|---|---|
| hopper-random-expert-diverse-10%-v2 | 57.1 | 64.9 |
| walker2d-random-expert-diverse-10%-v2 | 1.4 | 8.2 |

Table 3: Reweighting all terms in Equation 17 shows better performance than reweighting only the regularization term. See Section A.5.5.

| | | Uniform | AW | PF | DW+AW (ours) | DW+Uniform (ours) |
|---|---|---|---|---|---|---|
| **D4RL MuJoCo** | hopper-random-v2 | 9.2 | 7.9 | 7.8 | 6.5 | 9.6 |
| | hopper-medium-expert-v2 | 104.5 | 104.6 | 107.8 | 104.9 | 92.9 |
| | hopper-medium-replay-v2 | 85.9 | 97.0 | 91.2 | 97.3 | 88.6 |
| | hopper-full-replay-v2 | 100.5 | 101.2 | 102.1 | 101.4 | 99.7 |
| | hopper-medium-v2 | 62.1 | 67.7 | 65.7 | 65.2 | 66.1 |
| | hopper-expert-v2 | 107.2 | 108.2 | 108.6 | 108.0 | 105.4 |
| | halfcheetah-random-v2 | 21.4 | 16.3 | 3.0 | 11.1 | 12.5 |
| | halfcheetah-medium-expert-v2 | 71.2 | 89.4 | 73.4 | 88.8 | 86.1 |
| | halfcheetah-medium-replay-v2 | 45.3 | 44.7 | 42.2 | 44.3 | 45.1 |
| | halfcheetah-full-replay-v2 | 75.1 | 76.7 | 75.0 | 78.3 | 77.1 |
| | halfcheetah-medium-v2 | 46.5 | 46.5 | 45.4 | 46.6 | 46.5 |
| | halfcheetah-expert-v2 | 81.3 | 87.6 | 65.2 | 22.0 | 57.1 |
| | ant-random-v2 | 7.6 | 7.7 | 7.1 | 29.5 | 32.4 |
| | ant-medium-expert-v2 | 128.8 | 129.9 | 127.8 | 131.3 | 118.1 |
| | ant-medium-replay-v2 | 96.0 | 88.6 | 82.7 | 85.9 | 95.3 |
| | ant-full-replay-v2 | 129.3 | 124.5 | 127.5 | 128.4 | 129.1 |
| | ant-medium-v2 | 100.0 | 92.3 | 94.0 | 91.9 | 97.9 |
| | ant-expert-v2 | 124.5 | 132.0 | 130.0 | 127.2 | 129.6 |
| | walker2d-random-v2 | 6.1 | 4.8 | 13.2 | 7.8 | 8.6 |
| | walker2d-medium-expert-v2 | 109.6 | 109.3 | 108.9 | 109.4 | 109.7 |
| | walker2d-medium-replay-v2 | 74.4 | 78.1 | 71.9 | 78.4 | 75.3 |
| | walker2d-full-replay-v2 | 91.2 | 88.5 | 90.5 | 92.7 | 90.8 |
| | walker2d-medium-v2 | 82.3 | 81.3 | 78.2 | 81.9 | 82.1 |
| | walker2d-expert-v2 | 108.9 | 108.5 | 109.0 | 108.8 | 108.1 |
| **D4RL Antmaze** | antmaze-umaze-v0 | 76.0 | 77.3 | 69.3 | 76.7 | 72.7 |
| | antmaze-umaze-diverse-v0 | 48.0 | 36.0 | 24.8 | 40.0 | 34.0 |
| | antmaze-medium-diverse-v0 | 0.0 | 6.0 | 0.0 | 5.3 | 4.0 |
| | antmaze-medium-play-v0 | 2.4 | 10.7 | 0.0 | 8.7 | 1.3 |
| | antmaze-large-diverse-v0 | 0.0 | 2.0 | 2.7 | 1.3 | 2.0 |
| | antmaze-large-play-v0 | 0.4 | 1.3 | 0.0 | 0.7 | 0.0 |
| **D4RL Kitchen** | kitchen-complete-v0 | 27.8 | 30.2 | 9.5 | 9.5 | 16.5 |
| | kitchen-partial-v0 | 45.0 | 36.0 | 50.8 | 54.0 | 30.0 |
| | kitchen-mixed-v0 | 41.5 | 50.5 | 52.0 | 47.5 | 43.8 |
| **D4RL Adroit** | pen-human-v1 | 1.6 | -3.0 | 2.6 | -4.6 | 46.9 |
| | pen-cloned-v1 | -1.3 | -2.5 | 8.6 | -3.6 | 9.1 |
| | hammer-human-v1 | -7.0 | -7.0 | -6.9 | -7.0 | -7.0 |
| | hammer-cloned-v1 | -7.0 | -7.0 | -7.0 | -6.9 | -6.9 |
| | door-human-v1 | -9.4 | -9.4 | -5.1 | -9.4 | -9.4 |
| | door-cloned-v1 | -9.4 | -9.4 | 21.6 | -9.4 | -9.4 |
| | relocate-human-v1 | 2.1 | -2.1 | -0.8 | -2.1 | -2.1 |
| | relocate-cloned-v1 | -2.3 | -2.4 | 0.2 | -2.1 | -2.0 |
| **Mixed** | ant-random-medium-1%-v2 | 37.1 | 73.6 | 6.7 | 71.5 | 62.1 |
| | ant-random-medium-5%-v2 | 53.1 | 86.1 | 76.6 | 90.2 | 90.0 |
| | ant-random-medium-10%-v2 | 82.8 | 88.1 | 93.1 | 90.3 | 86.7 |
| | ant-random-medium-50%-v2 | 97.4 | 94.5 | 95.7 | 97.4 | 97.2 |
| | ant-random-expert-1%-v2 | 10.0 | 77.7 | 5.0 | 66.1 | 42.5 |
| | ant-random-expert-5%-v2 | 35.2 | 114.8 | 65.0 | 115.5 | 96.3 |
| | ant-random-expert-10%-v2 | 48.3 | 120.0 | 110.0 | 126.3 | 110.4 |
| | ant-random-expert-50%-v2 | 117.3 | 130.6 | 125.5 | 132.7 | 115.6 |
| | hopper-random-medium-1%-v2 | 0.6 | 55.1 | 62.2 | 57.2 | 68.7 |
| | hopper-random-medium-5%-v2 | 1.5 | 62.1 | 42.6 | 63.1 | 63.0 |
| | hopper-random-medium-10%-v2 | 1.6 | 66.6 | 66.8 | 69.2 | 60.7 |
| | hopper-random-medium-50%-v2 | 22.6 | 46.7 | 66.9 | 64.4 | 64.5 |
| | hopper-random-expert-1%-v2 | 17.5 | 59.6 | 17.4 | 86.1 | 21.3 |
| | hopper-random-expert-5%-v2 | 16.3 | 99.7 | 40.1 | 107.3 | 59.7 |
| | hopper-random-expert-10%-v2 | 14.9 | 109.7 | 46.6 | 109.3 | 55.1 |
| | hopper-random-expert-50%-v2 | 100.6 | 109.6 | 108.5 | 106.7 | 84.2 |
| | halfcheetah-random-medium-1%-v2 | 37.1 | 39.8 | 18.9 | 39.6 | 36.9 |
| | halfcheetah-random-medium-5%-v2 | 41.1 | 45.4 | 42.6 | 45.4 | 43.9 |
| | halfcheetah-random-medium-10%-v2 | 44.6 | 45.8 | 45.0 | 45.8 | 42.2 |
| | halfcheetah-random-medium-50%-v2 | 46.6 | 46.5 | 45.1 | 46.5 | 46.5 |
| | halfcheetah-random-expert-1%-v2 | 21.4 | 26.4 | 5.1 | 18.9 | 18.8 |
| | halfcheetah-random-expert-5%-v2 | 24.8 | 66.2 | 7.9 | 69.7 | 20.6 |
| | halfcheetah-random-expert-10%-v2 | 31.7 | 72.6 | 75.4 | 77.9 | 29.3 |
| | halfcheetah-random-expert-50%-v2 | 58.7 | 80.7 | 61.5 | 86.3 | 52.8 |
| | walker2d-random-medium-1%-v2 | 2.9 | 41.9 | 3.3 | 51.5 | 0.8 |
| | walker2d-random-medium-5%-v2 | 0.0 | 75.0 | 46.8 | 75.2 | 15.9 |
| | walker2d-random-medium-10%-v2 | 0.6 | 74.6 | 74.0 | 79.9 | 69.2 |
| | walker2d-random-medium-50%-v2 | 76.9 | 82.0 | 82.1 | 81.1 | 79.5 |
| | walker2d-random-expert-1%-v2 | 4.0 | 66.3 | 5.7 | 89.1 | -0.1 |
| | walker2d-random-expert-5%-v2 | 0.2 | 107.7 | 32.7 | 108.3 | 21.3 |
| | walker2d-random-expert-10%-v2 | 3.1 | 108.1 | 34.3 | 108.6 | 0.8 |
| | walker2d-random-expert-50%-v2 | 0.8 | 108.6 | 108.2 | 108.8 | 96.2 |
| **Mixed (diverse)** | ant-random-medium-diverse-1%-v2 | 9.6 | 20.7 | 5.9 | 74.1 | 35.5 |
| | ant-random-medium-diverse-5%-v2 | 53.3 | 85.4 | 39.5 | 85.1 | 78.4 |
| | ant-random-medium-diverse-10%-v2 | 78.0 | 89.9 | 93.2 | 95.7 | 93.8 |
| | ant-random-medium-diverse-50%-v2 | 101.9 | 91.3 | 93.7 | 97.7 | 102.4 |
| | ant-random-expert-diverse-1%-v2 | 7.5 | 9.8 | 6.2 | 33.5 | 13.6 |
| | ant-random-expert-diverse-5%-v2 | 12.7 | 29.1 | 8.2 | 102.0 | 66.8 |
| | ant-random-expert-diverse-10%-v2 | 23.4 | 76.5 | 58.1 | 113.7 | 89.7 |
| | ant-random-expert-diverse-50%-v2 | 112.5 | 116.3 | 121.3 | 126.8 | 123.9 |
| | hopper-random-medium-diverse-1%-v2 | 3.9 | 21.1 | 52.5 | 61.1 | 13.8 |
| | hopper-random-medium-diverse-5%-v2 | 24.5 | 24.5 | 55.5 | 65.7 | 73.0 |
| | hopper-random-medium-diverse-10%-v2 | 7.2 | 7.9 | 68.8 | 65.3 | 64.8 |
| | hopper-random-medium-diverse-50%-v2 | 32.4 | 62.0 | 58.7 | 61.5 | 63.2 |
| | hopper-random-expert-diverse-1%-v2 | 12.0 | 20.0 | 3.6 | 21.1 | 4.3 |
| | hopper-random-expert-diverse-5%-v2 | 5.2 | 10.3 | 27.7 | 60.0 | 37.0 |
| | hopper-random-expert-diverse-10%-v2 | 5.4 | 61.1 | 24.5 | 60.9 | 80.2 |
| | hopper-random-expert-diverse-50%-v2 | 96.8 | 107.4 | 92.3 | 109.2 | 105.5 |
| | halfcheetah-random-medium-diverse-1%-v2 | 39.6 | 41.2 | 32.7 | 26.0 | 41.1 |
| | halfcheetah-random-medium-diverse-5%-v2 | 44.5 | 44.2 | 44.7 | 39.8 | 45.5 |
| | halfcheetah-random-medium-diverse-10%-v2 | 43.7 | 45.3 | 44.9 | 45.2 | 46.2 |
| | halfcheetah-random-medium-diverse-50%-v2 | 46.8 | 46.8 | 45.7 | 45.4 | 46.5 |
| | halfcheetah-random-expert-diverse-1%-v2 | 11.8 | 22.0 | 7.5 | 2.9 | 17.0 |
| | halfcheetah-random-expert-diverse-5%-v2 | 29.4 | 35.4 | 10.6 | 7.0 | 25.7 |
| | halfcheetah-random-expert-diverse-10%-v2 | 24.1 | 39.5 | 20.3 | 16.4 | 18.7 |
| | halfcheetah-random-expert-diverse-50%-v2 | 53.3 | 69.1 | 8.9 | 52.2 | 71.2 |
| | walker2d-random-medium-diverse-1%-v2 | 2.6 | 2.5 | 26.8 | 3.1 | 3.0 |
| | walker2d-random-medium-diverse-5%-v2 | 0.7 | 8.4 | 47.3 | 52.1 | 18.0 |
| | walker2d-random-medium-diverse-10%-v2 | 0.6 | 24.7 | 57.8 | 75.5 | 71.7 |
| | walker2d-random-medium-diverse-50%-v2 | 76.9 | 80.0 | 77.5 | 78.4 | 79.0 |
| | walker2d-random-expert-diverse-1%-v2 | 12.3 | 6.1 | 0.0 | 9.1 | 9.7 |
| | walker2d-random-expert-diverse-5%-v2 | 1.1 | 3.7 | 85.3 | 0.7 | 5.0 |
| | walker2d-random-expert-diverse-10%-v2 | 1.7 | 3.2 | 49.9 | 0.9 | 33.9 |
| | walker2d-random-expert-diverse-50%-v2 | 7.9 | 1.7 | 79.2 | 95.9 | 108.3 |
| **Mixed (small)** | ant-random-medium-10%-small-v2 | 6.2 | 29.9 | 6.7 | 19.0 | 43.9 |
| | ant-random-expert-10%-small-v2 | 6.2 | 34.9 | 6.8 | 11.8 | 20.7 |
| | hopper-random-medium-10%-small-v2 | 39.7 | 4.8 | 46.0 | 55.6 | 51.5 |
| | hopper-random-expert-10%-small-v2 | 20.4 | 10.1 | 18.4 | 56.7 | 47.2 |
| | halfcheetah-random-medium-10%-small-v2 | 11.2 | 24.6 | 30.8 | 25.6 | 25.4 |
| | halfcheetah-random-expert-10%-small-v2 | 2.2 | 3.5 | 3.2 | 4.5 | 4.0 |
| | walker2d-random-medium-10%-small-v2 | 2.2 | 0.7 | 0.6 | 41.2 | 42.7 |
| | walker2d-random-expert-10%-small-v2 | 13.6 | -0.0 | 0.2 | 38.5 | 56.7 |

Table 4: Full results of average returns of CQL in total of 113 datasets.

| | | Uniform | AW | PF | DW+AW (ours) | DW+Uniform (ours) |
|---|---|---|---|---|---|---|
| D4RL MuJoCo | hopper-random-v2 | 7.6 | 6.8 | 8.0 | 6.4 | 8.5 |
| | hopper-medium-expert-v2 | 85.4 | 111.1 | 111.8 | 110.8 | 81.0 |
| | hopper-medium-replay-v2 | 86.7 | 98.1 | 96.0 | 99.9 | 79.7 |
| | hopper-full-replay-v2 | 108.1 | 102.1 | 88.2 | 107.5 | 99.8 |
| | hopper-medium-v2 | 65.7 | 58.3 | 64.5 | 61.7 | 62.5 |
| | hopper-expert-v2 | 109.7 | 111.0 | 110.3 | 105.7 | 108.2 |
| | halfcheetah-random-v2 | 12.7 | 7.3 | 4.2 | 10.7 | 10.8 |
| | halfcheetah-medium-expert-v2 | 90.6 | 94.7 | 94.2 | 93.9 | 93.7 |
| | halfcheetah-medium-replay-v2 | 44.0 | 44.0 | 29.4 | 44.1 | 44.6 |
| | halfcheetah-full-replay-v2 | 73.5 | 76.3 | 72.3 | 76.4 | 75.9 |
| | halfcheetah-medium-v2 | 47.5 | 47.8 | 45.4 | 47.9 | 47.7 |
| | halfcheetah-expert-v2 | 94.9 | 95.3 | 73.8 | 95.2 | 95.1 |
| | ant-v2 | 11.9 | 12.2 | 8.3 | 15.8 | 16.3 |
| | ant-medium-expert-v2 | 133.3 | 131.9 | 133.2 | 129.3 | 130.1 |
| | ant-medium-replay-v2 | 93.8 | 82.9 | 71.4 | 86.8 | 89.8 |
| | ant-full-replay-v2 | 130.1 | 129.9 | 128.9 | 131.4 | 130.2 |
| | ant-medium-v2 | 100.0 | 98.9 | 96.2 | 98.1 | 99.6 |
| | ant-expert-v2 | 126.2 | 131.4 | 119.6 | 128.6 | 127.3 |
| | walker2d-random-v2 | 6.7 | 2.7 | 10.4 | 3.7 | 7.0 |
| | walker2d-medium-expert-v2 | 110.1 | 109.7 | 109.8 | 109.8 | 109.7 |
| | walker2d-medium-replay-v2 | 61.3 | 47.0 | 42.2 | 62.6 | 65.1 |
| | walker2d-full-replay-v2 | 86.8 | 84.5 | 85.6 | 80.6 | 95.0 |
| | walker2d-medium-v2 | 77.9 | 70.0 | 65.3 | 75.8 | 80.8 |
| | walker2d-expert-v2 | 109.9 | 109.9 | 109.6 | 109.5 | 109.4 |
| D4RL Antmaze | antmaze-umaze-v0 | 88.0 | 90.7 | 0.0 | 89.3 | 81.3 |
| | antmaze-umaze-diverse-v0 | 67.3 | 75.3 | 0.0 | 72.0 | 61.0 |
| | antmaze-medium-diverse-v0 | 76.0 | 61.3 | 0.0 | 70.0 | 78.7 |
| | antmaze-medium-play-v0 | 72.0 | 22.0 | 0.0 | 30.0 | 64.7 |
| | antmaze-large-diverse-v0 | 36.7 | 23.3 | 0.0 | 20.7 | 40.0 |
| | antmaze-large-play-v0 | 43.3 | 9.3 | 0.0 | 10.0 | 42.0 |
| D4RL Kitchen | kitchen-complete-v0 | 62.8 | 26.3 | 10.0 | 19.8 | 60.0 |
| | kitchen-partial-v0 | 47.7 | 73.2 | 72.3 | 66.3 | 57.0 |
| | kitchen-mixed-v0 | 49.8 | 47.8 | 52.2 | 24.3 | 36.7 |
| D4RL Adroit | pen-human-v1 | 80.4 | 83.2 | 36.3 | 88.3 | 74.9 |
| | pen-cloned-v1 | 82.9 | 89.2 | 53.8 | 84.4 | 91.5 |
| | hammer-human-v1 | 3.1 | 0.5 | 3.2 | 0.8 | 1.2 |
| | hammer-cloned-v1 | 1.1 | 1.4 | 1.0 | 2.3 | 1.4 |
| | door-human-v1 | 2.5 | 0.6 | 0.1 | 0.0 | 1.4 |
| | door-cloned-v1 | 0.0 | 0.6 | 2.4 | -0.0 | 1.5 |
| | relocate-human-v1 | 0.5 | 0.0 | -0.0 | -0.0 | 0.1 |
| | relocate-cloned-v1 | -0.0 | 0.1 | 0.0 | 0.0 | -0.0 |
| Mixed | ant-random-medium-1%-v2 | 17.5 | 56.0 | 5.1 | 58.5 | 55.3 |
| | ant-random-medium-5%-v2 | 68.1 | 83.3 | 15.4 | 87.6 | 89.3 |
| | ant-random-medium-10%-v2 | 82.0 | 88.8 | 40.2 | 91.3 | 88.6 |
| | ant-random-medium-50%-v2 | 93.7 | 101.4 | 96.8 | 94.3 | 98.9 |
| | ant-random-expert-1%-v2 | 13.7 | 28.5 | 5.5 | 31.5 | 43.5 |
| | ant-random-expert-5%-v2 | 36.3 | 100.9 | 5.5 | 95.2 | 105.4 |
| | ant-random-expert-10%-v2 | 73.7 | 126.0 | 14.0 | 125.4 | 115.0 |
| | ant-random-expert-50%-v2 | 122.5 | 128.2 | 127.7 | 130.3 | 125.4 |
| | hopper-random-medium-1%-v2 | 52.2 | 56.1 | 42.4 | 56.5 | 51.7 |
| | hopper-random-medium-5%-v2 | 59.0 | 57.1 | 63.4 | 46.2 | 59.8 |
| | hopper-random-medium-10%-v2 | 63.2 | 57.1 | 65.3 | 63.6 | 62.8 |
| | hopper-random-medium-50%-v2 | 50.6 | 56.2 | 57.2 | 61.2 | 61.9 |
| | hopper-random-expert-1%-v2 | 11.1 | 74.8 | 16.4 | 64.8 | 22.2 |
| | hopper-random-expert-5%-v2 | 22.7 | 111.3 | 24.9 | 110.0 | 22.4 |
| | hopper-random-expert-10%-v2 | 46.7 | 111.5 | 33.9 | 110.6 | 64.3 |
| | hopper-random-expert-50%-v2 | 88.0 | 111.7 | 92.2 | 109.4 | 105.1 |
| | halfcheetah-random-medium-1%-v2 | 31.0 | 13.9 | 3.0 | 22.0 | 7.2 |
| | halfcheetah-random-medium-5%-v2 | 39.1 | 41.7 | 25.9 | 42.2 | 11.6 |
| | halfcheetah-random-medium-10%-v2 | 40.3 | 43.1 | 45.3 | 45.0 | 45.1 |
| | halfcheetah-random-medium-50%-v2 | 45.3 | 47.3 | 43.4 | 47.1 | 46.6 |
| | halfcheetah-random-expert-1%-v2 | 4.2 | 3.8 | 2.3 | 2.8 | 3.6 |
| | halfcheetah-random-expert-5%-v2 | 9.1 | 74.0 | 4.4 | 48.7 | 55.4 |
| | halfcheetah-random-expert-10%-v2 | 17.0 | 91.3 | 81.5 | 87.1 | 70.6 |
| | halfcheetah-random-expert-50%-v2 | 83.7 | 94.8 | 32.0 | 94.4 | 93.8 |
| | walker2d-random-medium-1%-v2 | 54.6 | 45.4 | 39.4 | 49.4 | 61.8 |
| | walker2d-random-medium-5%-v2 | 66.2 | 62.8 | 47.3 | 67.8 | 67.9 |
| | walker2d-random-medium-10%-v2 | 63.4 | 65.8 | 62.6 | 62.8 | 74.3 |
| | walker2d-random-medium-50%-v2 | 70.7 | 70.0 | 69.6 | 75.6 | 74.1 |
| | walker2d-random-expert-1%-v2 | 20.0 | 9.6 | 11.4 | 9.8 | 35.9 |
| | walker2d-random-expert-5%-v2 | 25.3 | 108.6 | 93.5 | 104.3 | 65.0 |
| | walker2d-random-expert-10%-v2 | 64.4 | 109.3 | 107.2 | 109.1 | 58.1 |
| | walker2d-random-expert-50%-v2 | 109.2 | 109.4 | 109.6 | 109.3 | 109.4 |
| Mixed (diverse) | ant-random-medium-diverse-1%-v2 | 12.6 | 29.4 | 11.7 | 40.8 | 24.1 |
| | ant-random-medium-diverse-5%-v2 | 29.1 | 83.7 | 24.8 | 88.1 | 73.3 |
| | ant-random-medium-diverse-10%-v2 | 61.0 | 91.8 | 54.6 | 89.3 | 89.3 |
| | ant-random-medium-diverse-50%-v2 | 91.2 | 98.0 | 89.0 | 97.0 | 96.8 |
| | ant-random-expert-diverse-1%-v2 | 12.4 | 20.0 | 10.1 | 30.4 | 20.7 |
| | ant-random-expert-diverse-5%-v2 | 17.1 | 77.4 | 10.9 | 86.2 | 71.5 |
| | ant-random-expert-diverse-10%-v2 | 27.9 | 101.1 | 22.9 | 100.5 | 93.8 |
| | ant-random-expert-diverse-50%-v2 | 101.9 | 123.3 | 122.8 | 127.7 | 126.9 |
| | hopper-random-medium-diverse-1%-v2 | 37.0 | 48.2 | 47.4 | 54.9 | 63.0 |
| | hopper-random-medium-diverse-5%-v2 | 46.7 | 44.7 | 49.3 | 48.6 | 49.0 |
| | hopper-random-medium-diverse-10%-v2 | 47.6 | 47.8 | 47.2 | 51.9 | 53.2 |
| | hopper-random-medium-diverse-50%-v2 | 51.2 | 48.7 | 53.6 | 54.6 | 52.1 |
| | hopper-random-expert-diverse-1%-v2 | 10.5 | 14.6 | 6.5 | 22.3 | 17.4 |
| | hopper-random-expert-diverse-5%-v2 | 17.1 | 59.0 | 31.4 | 78.7 | 41.0 |
| | hopper-random-expert-diverse-10%-v2 | 33.7 | 92.2 | 52.3 | 103.1 | 63.0 |
| | hopper-random-expert-diverse-50%-v2 | 90.1 | 106.8 | 22.7 | 98.9 | 109.4 |
| | halfcheetah-random-medium-diverse-1%-v2 | 14.4 | 2.7 | 8.0 | 2.2 | 14.4 |
| | halfcheetah-random-medium-diverse-5%-v2 | 34.9 | 20.1 | 17.1 | 2.3 | 15.2 |
| | halfcheetah-random-medium-diverse-10%-v2 | 39.7 | 39.7 | 24.0 | 30.9 | 24.3 |
| | halfcheetah-random-medium-diverse-50%-v2 | 44.4 | 22.5 | 46.6 | 8.8 | 35.1 |
| | halfcheetah-random-expert-diverse-1%-v2 | 5.6 | 4.5 | 5.7 | 3.2 | 4.6 |
| | halfcheetah-random-expert-diverse-5%-v2 | 4.8 | 8.5 | 3.6 | 9.9 | 10.3 |
| | halfcheetah-random-expert-diverse-10%-v2 | 9.6 | 16.4 | 4.3 | 13.2 | 28.0 |
| | halfcheetah-random-expert-diverse-50%-v2 | 66.4 | 70.7 | -0.9 | 72.9 | 85.0 |
| | walker2d-random-medium-diverse-1%-v2 | 36.0 | 33.3 | 14.6 | 61.3 | 59.8 |
| | walker2d-random-medium-diverse-5%-v2 | 67.5 | 61.7 | 64.8 | 67.9 | 55.3 |
| | walker2d-random-medium-diverse-10%-v2 | 58.8 | 58.9 | 59.2 | 66.2 | 60.5 |
| | walker2d-random-medium-diverse-50%-v2 | 66.8 | 49.4 | 74.6 | 63.7 | 64.0 |
| | walker2d-random-expert-diverse-1%-v2 | 10.5 | 13.0 | 1.6 | 31.0 | 26.4 |
| | walker2d-random-expert-diverse-5%-v2 | 19.2 | 33.1 | 7.7 | 66.4 | 76.4 |
| | walker2d-random-expert-diverse-10%-v2 | 49.9 | 61.2 | 8.5 | 73.1 | 86.4 |
| | walker2d-random-expert-diverse-50%-v2 | 64.1 | 108.8 | 15.3 | 108.1 | 108.6 |
| Mixed (small) | ant-random-medium-10%-small-v2 | 8.5 | 53.2 | 4.5 | 52.7 | 21.7 |
| | ant-random-expert-10%-small-v2 | 8.1 | 27.8 | 4.0 | 24.0 | 13.9 |
| | hopper-random-medium-10%-small-v2 | 11.0 | 51.7 | 9.5 | 49.5 | 39.0 |
| | hopper-random-expert-10%-small-v2 | 3.5 | 20.6 | 4.0 | 29.9 | 8.0 |
| | halfcheetah-random-medium-10%-small-v2 | 3.7 | 15.0 | 22.6 | 14.6 | 7.3 |
| | halfcheetah-random-expert-10%-small-v2 | 2.4 | 3.1 | -1.9 | 3.2 | 2.2 |
| | walker2d-random-medium-10%-small-v2 | 1.6 | 24.1 | 0.3 | 37.6 | 8.4 |
| | walker2d-random-expert-10%-small-v2 | 0.5 | 2.8 | 0.2 | 4.8 | 1.7 |

Table 5: Full results of average returns of IQL in total of 113 datasets.

|  |  | Uniform | AW | PF | DW+AW (ours) | DW+Uniform (ours) |
|---|---|---|---|---|---|---|
| **D4RL MuJoCo** | hopper-random-v2 | 8.5 | 9.0 | 7.5 | 6.4 | 8.5 |
|  | hopper-medium-expert-v2 | 95.4 | 105.6 | 106.5 | 108.8 | 97.5 |
|  | hopper-medium-replay-v2 | 64.2 | 96.8 | 83.4 | 89.3 | 56.2 |
|  | hopper-full-replay-v2 | 70.4 | 105.6 | 102.9 | 103.3 | 75.5 |
|  | hopper-medium-v2 | 59.8 | 63.8 | 65.2 | 60.6 | 57.9 |
|  | hopper-expert-v2 | 110.5 | 111.5 | 101.5 | 110.0 | 107.5 |
|  | halfcheetah-random-v2 | 12.3 | 11.3 | 10.3 | 10.1 | 12.0 |
|  | halfcheetah-medium-expert-v2 | 88.9 | 97.7 | 88.0 | 96.8 | 96.0 |
|  | halfcheetah-medium-replay-v2 | 44.7 | 45.1 | 29.6 | 45.0 | 44.9 |
|  | halfcheetah-full-replay-v2 | 74.1 | 77.7 | 75.4 | 75.7 | 75.2 |
|  | halfcheetah-medium-v2 | 48.4 | 48.6 | 48.3 | 48.0 | 48.3 |
|  | halfcheetah-expert-v2 | 96.4 | 97.5 | 78.4 | 97.4 | 97.8 |
|  | ant-random-v2 | 35.2 | 11.5 | -0.1 | 43.8 | 37.6 |
|  | ant-medium-expert-v2 | 113.2 | 135.5 | 121.4 | 130.5 | 120.2 |
|  | ant-medium-replay-v2 | 104.1 | 100.9 | 50.0 | 109.2 | 80.3 |
|  | ant-full-replay-v2 | 136.3 | 139.7 | 131.1 | 139.9 | 134.2 |
|  | ant-medium-v2 | 122.9 | 120.1 | 95.3 | 115.0 | 119.0 |
|  | ant-expert-v2 | 105.4 | 124.9 | 94.4 | 123.1 | 116.7 |
|  | walker2d-random-v2 | 1.2 | 2.5 | 2.5 | 5.3 | 2.6 |
|  | walker2d-medium-expert-v2 | 110.0 | 110.2 | 110.5 | 110.3 | 110.0 |
|  | walker2d-medium-replay-v2 | 80.2 | 80.8 | 72.4 | 81.1 | 77.4 |
|  | walker2d-full-replay-v2 | 93.3 | 96.2 | 96.0 | 95.7 | 93.6 |
|  | walker2d-medium-v2 | 84.4 | 82.3 | 76.0 | 82.6 | 83.2 |
|  | walker2d-expert-v2 | 110.2 | 110.3 | 110.2 | 110.3 | 110.2 |
| **D4RL Antmaze** | antmaze-umaze-v0 | 17.3 | 32.3 | 53.7 | 46.3 | 85.0 |
|  | antmaze-umaze-diverse-v0 | 64.7 | 67.3 | 29.0 | 66.3 | 40.3 |
|  | antmaze-medium-diverse-v0 | 3.7 | 10.7 | 3.3 | 4.0 | 1.0 |
|  | antmaze-medium-play-v0 | 0.0 | 1.0 | 2.0 | 0.0 | 0.3 |
|  | antmaze-large-diverse-v0 | 0.0 | 0.3 | 0.0 | 3.0 | 0.0 |
|  | antmaze-large-play-v0 | 0.0 | 0.0 | 0.0 | 0.0 | 0.0 |
| **D4RL Kitchen** | kitchen-complete-v0 | 0.0 | 0.0 | 0.0 | 0.0 | 0.0 |
|  | kitchen-partial-v0 | 0.0 | 0.8 | 0.0 | 9.6 | 17.8 |
|  | kitchen-mixed-v0 | 0.0 | 9.8 | 1.9 | 24.5 | 17.0 |
| **D4RL Adroit** | pen-human-v1 | 3.1 | 1.5 | 8.8 | -2.5 | 29.0 |
|  | pen-cloned-v1 | 11.9 | 2.2 | 17.1 | -1.1 | 33.4 |
|  | hammer-human-v1 | 1.1 | 0.8 | 0.4 | 1.2 | 1.3 |
|  | hammer-cloned-v1 | 0.2 | 0.4 | 0.4 | 0.3 | 0.2 |
|  | door-human-v1 | -0.3 | -0.3 | -0.3 | -0.3 | -0.3 |
|  | door-cloned-v1 | -0.3 | -0.3 | -0.3 | -0.3 | -0.3 |
|  | relocate-human-v1 | -0.3 | -0.3 | -0.3 | -0.3 | -0.3 |
|  | relocate-cloned-v1 | -0.3 | -0.2 | -0.3 | -0.4 | -0.4 |
| **Mixed** | ant-random-medium-1%-v2 | 41.2 | 21.6 | -0.5 | 21.6 | 47.2 |
|  | ant-random-medium-5%-v2 | 41.9 | 84.2 | 0.9 | 76.1 | 68.8 |
|  | ant-random-medium-10%-v2 | 55.6 | 84.5 | 29.2 | 84.6 | 75.2 |
|  | ant-random-medium-50%-v2 | 85.2 | 114.5 | 111.5 | 120.0 | 108.2 |
|  | ant-random-expert-1%-v2 | 18.7 | 14.0 | 0.3 | 16.6 | 16.7 |
|  | ant-random-expert-5%-v2 | 27.8 | 39.6 | 4.8 | 47.6 | 71.0 |
|  | ant-random-expert-10%-v2 | 44.0 | 65.9 | 13.6 | 83.5 | 79.5 |
|  | ant-random-expert-50%-v2 | 31.6 | 97.8 | 96.3 | 117.7 | 63.2 |
|  | hopper-random-medium-1%-v2 | 25.7 | 49.1 | 13.4 | 51.3 | 20.3 |
|  | hopper-random-medium-5%-v2 | 41.2 | 58.0 | 49.8 | 51.7 | 45.8 |
|  | hopper-random-medium-10%-v2 | 29.4 | 56.4 | 50.9 | 53.8 | 55.9 |
|  | hopper-random-medium-50%-v2 | 55.8 | 64.7 | 53.7 | 61.1 | 51.9 |
|  | hopper-random-expert-1%-v2 | 21.3 | 36.3 | 10.5 | 17.3 | 26.7 |
|  | hopper-random-expert-5%-v2 | 31.0 | 97.3 | 37.6 | 93.3 | 67.6 |
|  | hopper-random-expert-10%-v2 | 52.0 | 107.7 | 84.4 | 91.3 | 95.4 |
|  | hopper-random-expert-50%-v2 | 87.2 | 106.5 | 107.9 | 110.3 | 108.3 |
|  | halfcheetah-random-medium-1%-v2 | 15.8 | 14.8 | 27.4 | 16.5 | 40.9 |
|  | halfcheetah-random-medium-5%-v2 | 21.0 | 46.9 | 44.0 | 46.8 | 45.8 |
|  | halfcheetah-random-medium-10%-v2 | 36.2 | 47.8 | 47.8 | 48.3 | 46.8 |
|  | halfcheetah-random-medium-50%-v2 | 48.5 | 48.3 | 48.0 | 48.7 | 48.2 |
|  | halfcheetah-random-expert-1%-v2 | 2.7 | 3.6 | 8.2 | 3.1 | 14.3 |
|  | halfcheetah-random-expert-5%-v2 | 20.6 | 50.2 | 5.7 | 44.8 | 58.8 |
|  | halfcheetah-random-expert-10%-v2 | 25.9 | 78.4 | 82.6 | 70.9 | 69.4 |
|  | halfcheetah-random-expert-50%-v2 | 85.8 | 96.0 | 70.0 | 95.2 | 91.8 |
|  | walker2d-random-medium-1%-v2 | 6.0 | -0.2 | 7.8 | 29.9 | 5.2 |
|  | walker2d-random-medium-5%-v2 | 14.5 | 74.7 | 1.9 | 70.5 | 1.7 |
|  | walker2d-random-medium-10%-v2 | 9.9 | 74.2 | 2.3 | 75.3 | 3.6 |
|  | walker2d-random-medium-50%-v2 | 21.9 | 78.2 | 14.7 | 78.5 | 10.8 |
|  | walker2d-random-expert-1%-v2 | 5.3 | 15.4 | 0.3 | 4.2 | -0.3 |
|  | walker2d-random-expert-5%-v2 | 6.4 | 73.1 | 3.5 | 109.9 | 1.9 |
|  | walker2d-random-expert-10%-v2 | 3.0 | 110.1 | 1.5 | 110.0 | 2.7 |
|  | walker2d-random-expert-50%-v2 | 8.7 | 110.3 | 0.8 | 110.1 | 2.4 |
| **Mixed (diverse)** | ant-random-medium-diverse-1%-v2 | 40.5 | 43.9 | 5.7 | 84.3 | 64.1 |
|  | ant-random-medium-diverse-5%-v2 | 32.6 | 76.8 | 3.4 | 100.8 | 97.1 |
|  | ant-random-medium-diverse-10%-v2 | 44.5 | 105.3 | 3.8 | 96.7 | 48.6 |
|  | ant-random-medium-diverse-50%-v2 | 95.7 | 118.3 | 110.8 | 115.8 | 107.2 |
|  | ant-random-expert-diverse-1%-v2 | 23.7 | 24.5 | 6.0 | 30.3 | 22.2 |
|  | ant-random-expert-diverse-5%-v2 | 23.5 | 63.6 | 1.1 | 59.0 | 69.9 |
|  | ant-random-expert-diverse-10%-v2 | 28.9 | 76.6 | 3.4 | 85.0 | 81.2 |
|  | ant-random-expert-diverse-50%-v2 | 28.2 | 69.6 | 63.9 | 87.0 | 76.0 |
|  | hopper-random-medium-diverse-1%-v2 | 25.9 | 16.2 | 24.4 | 37.1 | 8.9 |
|  | hopper-random-medium-diverse-5%-v2 | 38.5 | 43.7 | 49.3 | 37.5 | 37.3 |
|  | hopper-random-medium-diverse-10%-v2 | 29.7 | 30.0 | 36.7 | 43.9 | 41.5 |
|  | hopper-random-medium-diverse-50%-v2 | 54.4 | 50.3 | 50.9 | 48.4 | 47.2 |
|  | hopper-random-expert-diverse-1%-v2 | 15.2 | 13.7 | 13.2 | 32.6 | 10.3 |
|  | hopper-random-expert-diverse-5%-v2 | 11.7 | 57.6 | 37.7 | 84.8 | 58.1 |
|  | hopper-random-expert-diverse-10%-v2 | 48.2 | 76.6 | 75.5 | 41.2 | 86.6 |
|  | hopper-random-expert-diverse-50%-v2 | 90.9 | 95.3 | 40.8 | 109.2 | 75.4 |
|  | halfcheetah-random-medium-diverse-1%-v2 | 39.5 | 15.1 | 27.7 | 33.5 | 39.7 |
|  | halfcheetah-random-medium-diverse-5%-v2 | 18.3 | 36.8 | 46.0 | 46.9 | 44.8 |
|  | halfcheetah-random-medium-diverse-10%-v2 | 20.4 | 23.2 | 46.6 | 47.4 | 45.5 |
|  | halfcheetah-random-medium-diverse-50%-v2 | 48.2 | 15.9 | 48.4 | 45.0 | 46.8 |
|  | halfcheetah-random-expert-diverse-1%-v2 | 3.4 | 2.3 | 5.7 | 10.2 | 11.9 |
|  | halfcheetah-random-expert-diverse-5%-v2 | 4.4 | 15.9 | 12.2 | 45.8 | 62.3 |
|  | halfcheetah-random-expert-diverse-10%-v2 | 6.3 | 13.5 | 12.3 | 64.6 | 80.2 |
|  | halfcheetah-random-expert-diverse-50%-v2 | 82.1 | 49.6 | 14.2 | 86.3 | 93.4 |
|  | walker2d-random-medium-diverse-1%-v2 | -0.4 | 5.1 | 7.9 | 2.1 | 8.9 |
|  | walker2d-random-medium-diverse-5%-v2 | 4.8 | 5.8 | 1.6 | 0.4 | 0.2 |
|  | walker2d-random-medium-diverse-10%-v2 | 4.7 | 10.6 | 15.0 | 4.5 | 0.3 |
|  | walker2d-random-medium-diverse-50%-v2 | 12.1 | 47.4 | 78.5 | 13.2 | 9.9 |
|  | walker2d-random-expert-diverse-1%-v2 | 4.4 | 2.1 | 1.5 | 5.5 | 2.2 |
|  | walker2d-random-expert-diverse-5%-v2 | 5.6 | 11.3 | 1.9 | 8.3 | -0.3 |
|  | walker2d-random-expert-diverse-10%-v2 | 5.5 | 4.5 | 1.0 | 7.6 | 8.2 |
|  | walker2d-random-expert-diverse-50%-v2 | 6.1 | 106.1 | 15.0 | 0.6 | 6.1 |
| **Mixed (small)** | ant-random-medium-10%-small-v2 | 9.3 | 20.0 | 4.3 | 15.8 | 17.2 |
|  | ant-random-expert-10%-small-v2 | 9.3 | 15.2 | 0.4 | 18.2 | 17.1 |
|  | hopper-random-medium-10%-small-v2 | 6.4 | 46.7 | 2.5 | 43.7 | 20.2 |
|  | hopper-random-expert-10%-small-v2 | 5.8 | 13.3 | 4.1 | 20.4 | 18.0 |
|  | halfcheetah-random-medium-10%-small-v2 | 21.5 | 11.9 | 6.5 | 4.0 | 23.2 |
|  | halfcheetah-random-expert-10%-small-v2 | 6.5 | 5.4 | -1.1 | 6.8 | 7.3 |
|  | walker2d-random-medium-10%-small-v2 | 1.8 | 12.9 | 0.9 | 6.4 | 6.3 |
|  | walker2d-random-expert-10%-small-v2 | 0.9 | -0.3 | 0.2 | 0.4 | 1.6 |

Table 6: Full results of average returns of TD3BC in total of 113 datasets.

