# OpenReview forum: "Beyond Uniform Sampling: Offline Reinforcement Learning with Imbalanced Datasets"
_NeurIPS.cc/2023/Conference — NeurIPS 2023 poster_

### Official Review · Reviewer_XiFQ · 2023-06-10

**Soundness:** 3 good
**Presentation:** 4 excellent
**Contribution:** 3 good
**Rating:** 6
**Confidence:** 4

**Summary:**

This paper investigates the offline RL with the imbalanced dataset setting. The proposed method, which optimizes a parameterized density-ratio weighting (DW) model for each transition, anchors a policy close to the good part of trajectories in the dataset. The proposed method is shown to outperform state-of-the-art methods on 72 imbalanced datasets both in varying imbalance and initial state distribution.

**Strengths:**

1. Overall good writing quality and clarity.
2. The proposed approach is well-motivated by adopting existing approaches.
3. Sufficient technical details for reproducing the experiments.

**Weaknesses:**

1. Novelty limited: Fixed conservativeness is defective for imbalanced (heterogeneous) datasets and reweighting or filtering samples/trajectories are not new in offline RL [1, 2, 3]. On the other hand, this paper does not connect with the previous most similar work [1] and explains why DW is better than AW, though it claims that AW is empirically prone to overfitting.
2. Lack of details for the dataset setup.
3. Lack of comparison: the imbalanced dataset is also called the noisy dataset [4]. If the authors follow the setup of the dataset from AW [1], the same as the noisy data setup [4], it is reasonable to include more comparisons between SQL/EQL enforcing a Q-function weighting on the transitions and show the edge of DW.
3. Experiments between AW and DW: In Figure 2, AW-M is better than DW-Uniform in three of four tasks. However, the authors claim that they are better than AW.  More explanation is needed about why DW is better than AW.



**Questions:**

4. It is reasonable to set $\gamma=1$ to avoid the estimation of $\rho_0$. Have the authors analyzed the role of the discount factor for offline RL [5]? It will be appreciated if any theoretical analysis or/and experiments are provided.
5. Lacking the AW-L benchmark in Figure 2(b). The medium temp is extremely beneficial in IQL (see Figure 2(b)), why do not present the performance of AW-L?
7. In Figure 3, It is counterintuitive that DW is not influenced by diverse initial states and varying trajectories because DW does not consider the initial state distribution. Could the authors explain this?
8. Thank the authors for providing technical details including their hyperparameter search range with $\lambda_f$ and $\lambda_k$. However, I am a bit concerned that the necessity of penalties, i.e., Bellman flow and KL-divergence. Could the authors please introduce more experiences to show their necessities and the guidance for hyperparameter research with new tasks?

[1] Hong Z W, Agrawal P, des Combes R T, et al. Harnessing Mixed Offline Reinforcement Learning Datasets via Trajectory Weighting[C]//The Eleventh International Conference on Learning Representations. 2023.

[2] Brandfonbrener D, Whitney W F, Ranganath R, et al. Quantile filtered imitation learning[J]. arXiv preprint arXiv:2112.00950, 2021.

[3] Wu Y, Zhai S, Srivastava N, et al. Uncertainty weighted actor-critic for offline reinforcement learning[J]. arXiv preprint arXiv:2105.08140, 2021.

[4] Xu H, Jiang L, Li J, et al. Offline rl with no ood actions: In-sample learning via implicit value regularization[J]. arXiv preprint arXiv:2303.15810, 2023.

[5] Hu H, Yang Y, Zhao Q, et al. On the role of discount factor in offline reinforcement learning[C]//International Conference on Machine Learning. PMLR, 2022: 9072-9098.



**Limitations:**

See questions and weaknesses.

---
raise the score to 6 after reviewing the response from the authors.

---

**Minors**:
1. Cite twice: The classic results in [34] indicate that a policy’s expected return can be expressed in terms of its stationary state-action distribution [34].
2. IQL training object for policy improvement (line 234) is wrong.
3. What is DW-uniform? Only use DW? I suggest changing the name of DW-uniform because *uniform* will cause misunderstanding that DW-uniform is a combination sampling method between DW and Uniform.

---

> ### Author Rebuttal · Authors · 2023-08-10
>
> We thank the reviewer for appreciating the quality and clarity of our paper, and breadth of our experimental evaluation. We respond to the main questions in the following (W=weakness, Q=questions).
>
> > W1: Novelty
>
> We're the first to use density ratio optimization to address fixed conservativeness in offline RL (like CQL, and IQL), differing from prior work [1, 2, 3].
>
> [1] need trajectories to start from similar initial states; otherwise, excessive weights can be assigned to trajectories starting from lucky initial states with higher returns, and data in other good trajectories with unlucky initial states will be underutilized. DW doesn't require trajectories to start from similar initial states, outperforming AW in datasets with diverse initial states in Fig. 3.
>
> [2] filter low-value data, but we maximize the expected rewards of the reweighted data distribution. Since their code is not released, we compared DW with its closet work [4], showing that our DW outperforms [4] in Fig. 7 in the attached PDF.
>
> [3] weight samples by Q-values' variance, but DW weights data by their rewards (with some constraints). In Fig. 7, UWAC underperforms the other methods, suggesting it may not be an effective approach to address the issues in imbalanced datasets.
>
> > W1: Why DW is better than AW
> >
>
> We want to clarify that we already cited and compared with AW [1], showing that DW-AW outperforms AW on various imbalanced datasets (Figs 2 and 3). **DW is better than AW as it can up-weight valuable data in both low- and high- return trajectories**, while AW only does this for high-return ones. This is crucial as valuable state-action pairs might be hidden in low-return trajectories. In Fig. 8 (PDF), AW and PF fail to find optimal state-action distribution due to missing optimal trajectories. In contrast, DW learns the optimal distribution, highlighting the benefit of DW over trajectory reweighting.
>
> > W2 & 3 and Q2: Lack of details for the dataset setup / Lack of comparison / Lack of AW-L
> >
>
> The dataset setup is in Appendix A.3.3. We’ve added AW-L, SQL and EQL in Fig. 7. AW-L is close to AW-M in small datasets. Both DW-AW and DW-Uniform outperform SQL and EQL in imbalanced datasets.
>
> > W4: Why DW is better than AW in Figure 2
> >
>
> DW is better than AW since combing DW and AW (i.e., DW-AW) achieves higher average return or match the baselines in Figures 2(a) and 2(b). Please let us know if this explanation answers the question.
>
> > Q1: Role of the discount factor for offline RL [5]
>
> We keep the original discount factors for training offline RL algorithms (CQL, IQL), only setting $\gamma=1$ for DW. The effects of discount factors in offline RL should directly transfer to the combination of DW with offline RL algorithms.
>
> > Q3: Why DW isn’t influenced by diverse initial states
>
> The reviewer might believe DW works only with a single initial state due to Eq. 10 omitting initial state distribution. However, we'd like to kindly clarify that DW's formulation isn't restricted to one initial state but agnostic to initial states because it optimizes undiscounted returns instead of discounted ones, as detailed in Sec. 4.1.
>
> > Q4: Necessity of Bellman flow and KL penalties.
>
> - **Bellman flow penalty is required** to ensure the learned weights are valid in the MDP (Sec. 4.1). Otherwise, maximizing the objective (Eq. 6) makes all the weights assigned to the data with the highest reward.
> - **KL penalty is required,** according to the theoretical analysis shown by Sec. 5.1 in [6]. Otherwise, high weights can be assigned to the actions that lead to next states that are absent in the dataset.
>
> > Q4: Guidance for hyperparameter search
>
> - We suggest starting with low **flow penalty $\lambda_F$** since it is lower bounded by zero, and increasing it often leads to performance gain. The following table shows the average return shown in Table 2 in Appendix, where K and F denote $\lambda_K$ and $\lambda_F$, respectively. Increasing F improves the performance in IQL. We found that CQL is less sensitive to F.
>
>     | K,F | 0.2,0.1 | 0.2,1.0 | 0.2,5.0 | 1.0,0.1 | 1.0,1.0 | 1.0,5.0 |
>     | --- | --- | --- | --- | --- | --- | --- |
>     | CQL | 48.1 | 41.7 | 47.5 | 10.1 | 7.5 | 11.5 |
>     | IQL | 32.0 | 56.0 | 52.4 | 47.2 | 58.5 | 61.5 |
> - Regarding **KL penalty weight $\lambda_K$**, we discuss in two datasets:
>     1. **Low-return dominant:** To ease offline RL's conservatism on data from low-performing policies, we suggest starting with low $\lambda_K$ since high $\lambda_K$ limits deviation from the dataset. The above table displays average performance in low-return dominant datasets, showing higher $\lambda_K$ can lead to drops in CQL.
>     2. **High-return dominant:** As data here is nearly optimal, we suggests beginning with high $\lambda_K$. The following table  shows the performance in **`halfcheetah-expert-v2`** dataset, showing that high $\lambda_K$ leads to better performance.
>
>   | K,F | 0.2,5.0 | 1.0,5.0 |
>   | --- | --- | --- |
>   | CQL | 59.2 | 81.9 |
>   | IQL | 95.2 | 95.1 |
>
> While optimal coefficients vary by dataset, we've shown that the same hyperparameters can enhance performance in imbalanced datasets while not harming overall performance in high-return dominant datasets from the original D4RL.
>
> > Minors
>
> Thank you for your feedback. We'll revise the manuscript accordingly. "DW-Uniform" refers to DW trained with uniform sampling, and "DW-AW" to DW trained with AW sampling, based on the optimization of Eq. 13.
>
> [1] Hong et al. “Harnessing Mixed Offline Reinforcement Learning Datasets via Trajectory Weighting”
>
> [2] Brandfonbrener et al. Quantile filtered imitation learning
>
> [3] Wu et al. Uncertainty weighted actor-critic for offline reinforcement learning
>
> [4] Chen et al. Bail: Best-action imitation learning for batch deep reinforcement learning
>
> [5] Hu et al. On the role of discount factor in offline reinforcement learning
>
> [6] Zhan et al. Offline reinforcement learning with realizability and single-policy concentrability

---

> > ### Comment · Reviewer_XiFQ · 2023-08-18
> > **Response to the Authors' Rebuttal by Reviewer XiFQ**
> >
> > I have carefully reviewed the comments from other reviewers, and the authors have addressed most of my concerns. The additional experiments provided sufficiently demonstrate the advantages of density ratio optimization. However, I still have one minor concern.
> >
> > Regarding my previous concerns about Q1 and Q3, the authors chose to set $\gamma=1$ to bypass the dependence on the initial state distribution, i.e., $\rho_0$. Nonetheless, I believe that setting $\gamma=1$, which is agnostic with the initial state distribution, comes with sacrifice. Could the authors discuss the limitations associated with this choice?
> >
> > Overall, I appreciate the detailed response and have decided to raise my rating to 6.

---

> > > ### Author Response · Authors · 2023-08-21
> > >
> > > We are glad that our response addresses the reviewer's concerns.
> > >
> > > **Regarding the choice to set $\gamma=1$:** This choice might not be aligned with the task objective when short-term rewards are preferable over long-term ones. For instance, in stock trading scenarios, one may prefer short-term revenues (rewards) over long-term ones since one may not be able to afford large losses before obtaining a huge revenue in the far future.
> > >
> > > We thank the reviewer again for the appreciation of our response and hope our follow-up response addresses the question.

---

### Official Review · Reviewer_qcFR · 2023-06-18

**Soundness:** 3 good
**Presentation:** 3 good
**Contribution:** 2 fair
**Rating:** 6
**Confidence:** 4

**Summary:**

Authors propose a new method for weighting the samples from datasets in order to mimic the sampling from dataset which is collected by the better policy the the  behavioral policy. This method can be integrated into any algorithm and combined with previous approaches.

**Strengths:**

Method seems to be not that hard to implement and it shouldn't introduce a notable computational overhead while can be applied with the arbitary algorithm.On imballanced locomotion datasets generated by the authors proposed approach helps to achieve much better results than other approaches. Benchmarking over high number of  the datasets is another  strength.

**Weaknesses:**

While approach helps to improve performance when D4RL locomotion datasets are mixed with the random datasets, it seems like there is no advantage  when applied to the original mixed datasets (medium-replay, medium-expert, full-replay) or the performance even drops on those if we look at Tables 4 and 5.


**Questions:**

What are the avereged scores across domains in tables 4 and 5? As mentioned in "Weaknesses" it seems like the proposed method does not benefit a lot (if benefit at all) when applied to the original D4RL datasets. Could you please add those numbers to see what is the situation on average? I mean average scores over locomotion, antmaze, kitchen, adroit, generated datasets and averaged scores over all of the domains.

How does the method affect algorithms time needed for training?

**Limitations:**

Beside the uncetainty whether approach helps when dataset is not imbalanced that hard there is another limitation in design. If I understand approach correctly, it can't be applied to the datasets which require trajectories stiching to complete task. The example of such task is AntMaze and authors' approach mostly decrease performance when applied to those datasets.

---

> ### Author Rebuttal · Authors · 2023-08-09
>
> We thank the reviewer for appreciating our extensive evaluation and simplicity of implementation. In the following, we answer each of the reviewer’s question.
> > Could you please add those numbers to see what is the situation on average?
> >
>
> **Answer:**
>
> **Advantage in the original mixed datasets:** DW performs similarly with AW because the original mixed datasets (i.e., medium-expert, medium-replay, full-replay) in D4RL already have enough proportion of good data. The mixed datasets used in our experiment (Figure 1) only have an average normalized trajectory return of 0.08, but the original mixed datasets in D4RL have 0.5, which is far higher than ours. As the average return of datasets indicates the performance of behavior policies, this means staying close to the behavior policy won’t hurt the performance much, which explains why reweighting doesn’t lead to much performance gain.
>
> **Average scores across domains:** Table 7 in the attached PDF shows the average scores for the original D4RL MuJoCo, antmaze, kitchen, and adroit datasets, as well as our imbalanced MuJoCo dataset. We also included the average scores across all domains. The numbers inside the parenthesis denote the relative score compared with the Uniform.
>
> **DW matches the baselines in the original D4RL datasets (D4RL Adroit, Antmaze, Kitchen, and Gym-MuJoCo).**
>
> - **For CQL**, both DW+AW and DW+Uniform match Uniform’s and AW’s performance, with average relative performance of +3.45 (DW+AW) and +1.47 (DW+Uniform). This shows that in terms of average performance, DW+AW and DW+Uniform even slightly improve the performance over Uniform in the original D4RL datasets.
> - **For IQL**, DW+AW and DW+Uniform also perform closely with the baselines, except for DW+AW in Antmaze and Kitchen. The performance drop of DW+AW is likely because AW is worse than Uniform by 16.9 and 5.9 points in both domains.
>
> > How does the method affect algorithms time needed for training?
> >
>
> **Answer:**
>
> We provide the running time of each method below. DW only incurs 10mins more in the training time, not causing excessive overhead.
> |  | Uniform | AW | PF | DW-AW | DW-Uniform |
> | --- | --- | --- | --- | --- | --- |
> | CQL | 58mins | 60mins | 60mins | 70mins | 67mins |
> | IQL | 40mins | 43mins | 43mins | 60mins | 57mins |
>
> > If I understand approach correctly, it can't be applied to the datasets which require trajectories stiching to complete task.
> >
> **Answer:**
>
> We would like to clarify that DW can be applied to tasks requiring trajectory stitching and illustrate why in the following.
>
> **DW doesn’t hurt in AntMaze.** The following table presents the average return of CQL and IQL with varying sampling/reweighting methods in AntMaze datasets. DW+Uniform matches the performance of Uniform, showing that it doesn’t hurt the performance in tasks requiring trajectory stitching. The reason why applying DW doesn’t improve over Uniform can be that offline RL algorithms have already done trajectory stitching well. Thus there is not much room for improving the sampling distribution. It is likely because the state-action distribution in AntMaze is not skewed toward low-performing trajectories. As such, regularized offline RL algorithms won’t suffer in such cases.
>
> |  | Uniform | AW | PF | DW+AW (ours) | DW+Uniform (ours) |
> | --- | --- | --- | --- | --- | --- |
> | CQL | 15.2 | 23.4 | 17.7 | 22.1 | 19.1 |
> | IQL | 63.8 | 47.0 | 0.0 | 45.5 | 63.2 |
>
> **Four-room experiment shows that DW can stitch trajectories.** To assess DW's trajectory stitching ability, we conducted an experiment in a didactic four-room environment [2].
>
> - **Experiment setup:** See Figure 8 in the attached PDF for the environment illustration. The agent starts from an orange initial state, traverses non-red cells, gaining +1 reward at the green goal, else zero. To test trajectory stitching, a suboptimal dataset with 1000 trajectories was generated, where none of each is optimal trajectory. Due to absence of optimal trajectories, up-weighting trajectories (i.e., AW and PF) with high-returns won’t produce a state-action distribution matching the optimal policy. Thus, if a method can generate state-action distribution matching the one of the optimal policy, it indicates that the method is able to stitch trajectories because it can identify optimal state-action pairs leading to the goal even though those optimal state-action pairs are not observed in the same trajectory in the dataset.
> - **Results:** Figures 8(a), 8(b), and 8(e) display the state-action distributions of behavior policy, optimal policy, and DW; the number above each plot is expected return under the state-action distribution. We see that both AW (Figure 8(c)) and PF (Figure (d)) fail to match the state-action distribution of the optimal policy in Figure 8(b), hence leading to suboptimal performance. In contrast, DW successfully approximates the optimal policy's state-action distribution, confirming DW can identify optimal state-action pairs observed in different suboptimal trajectories and stitch them to optimal trajectories.
>
> **DW improves BC in Ant U-maze.** To see if the learned importance weight mirrors a data distribution of a better policy than the behavior policy, we evaluate DW upon BC. This way, we can exclude trajectory stitching from offline RL algorithms like CQL and IQL since BC cannot do trajectory stitching. To improve beyond the behavior policy, one needs to change the data distribution to make it reflect a better policy. Our results show that DW achieves better performance than uniform sampling (65 v.s. 45), indicating that DW is applicable in tasks requiring trajectory stitching.
>
> [1] Hong Z W, Agrawal P, des Combes R T, et al. Harnessing Mixed Offline Reinforcement Learning Datasets via Trajectory Weighting, ICLR'23
>
> [2] Lee, Jongmin, et al. "Optidice: Offline policy optimization via stationary distribution correction estimation." ICML'21

---

> > ### Comment · Reviewer_qcFR · 2023-08-11
> >
> > Thank you very much for answering my questions and conducting additional experiments. I'm increasing the rating in my review.

---

> > > ### Author Response · Authors · 2023-08-17
> > >
> > > We are glad that our answers address the reviewer's questions. We will include these new experimental results in the updated manuscript.

---

### Official Review · Reviewer_yys4 · 2023-07-06

**Soundness:** 3 good
**Presentation:** 3 good
**Contribution:** 3 good
**Rating:** 7
**Confidence:** 4

**Summary:**

This paper proposes a method to improve offline reinforcement learning (RL) performance on imbalanced datasets, where most of the data comes from low-performing policies and only a few from high-performing ones. The method, called density-ratio weighting (DW), optimizes the importance sampling weights to emulate sampling data from a data distribution generated by a nearly optimal policy. The paper shows that DW can enhance the performance of state-of-the-art offline RL algorithms on 72 imbalanced datasets with varying types of imbalance.

**Strengths:**

- It proposes a novel method of DW that optimizes the importance sampling weights to emulate sampling data from a data distribution generated by a nearly optimal policy, rather than the original dataset.
- It demonstrates overall performance gains over SoTA offline RL algorithms and other baselines on imbalanced datasets with varying types and degrees of imbalance.
- Its written style is clear and easy to follow.

**Weaknesses:**

- The theoretical analysis is relatively weak.
- Slightly lack of discussion with a type of offline RL algorithms which filter out low-performance trajectories.

**Questions:**

- Though obtaining a performance bound is quite difficult as describing the imbalanced dataset is hard, is it possible to prove the optimization with the proposed loss function will converge? For example, I guess, derive an iteration equation of $w$ as the Bellman equation, and prove the operator is a contraction mapping.
- In Section 6 related work, the paper discusses offline imitation learning approaches. It claims that these approaches "assume prior knowledge of which data points are generated by experts". However, there are some offline imitation or RL methods which only use the reward signals in the dataset and filter out low-quality data to improve the final performance, such as BAIL [1] and COIL [2]. I am curious about what the advantages of DW are compared to them, especially for BAIL which does not need trajectory information. Also, it will be better if you can try BAIL on your imbalanced dataset and show the results.

[1] Chen, Xinyue, et al. "Bail: Best-action imitation learning for batch deep reinforcement learning." Advances in Neural Information Processing Systems 33 (2020): 18353-18363.

[2] Liu, Minghuan, et al. "Curriculum offline imitating learning." Advances in Neural Information Processing Systems 34 (2021): 6266-6277.

**Limitations:**

Yes.

---

> ### Author Rebuttal · Authors · 2023-08-09
>
> We thank the reviewer for appreciating the novelty of our work, extensive evaluation, and clarity of our writing. We clarify the reviewer’s questions in the following.
>
> > Though obtaining a performance bound is quite difficult as describing the imbalanced dataset is hard, is it possible to prove the optimization with the proposed loss function will converge? For example, I guess, derive an iteration equation of $w$ as the Bellman equation, and prove the operator is a contraction mapping.
> >
> **Answer:**
>
> Note that as we don’t use fixed point iteration (i.e., not using target network of $w$) like value iteration to learn $w$, the update operator doesn’t have to be contraction mapping for convergence. Regarding convergence, the optimization objective of $w$ is a convex optimization problem over $w(s,a)$. We use gradient descent to optimize w. According to the standard results in Boyd et al. [1], gradient descent converges as long as the optimization domain of $w$ is bounded, the gradient w.r.t., the loss $\nabla_w L(w)$, is bounded, and the learning rates are not too large asymptotically: $\sum_t \alpha_t^2 < \infty$. All three conditions are met in our case. Thus optimization of w will converge too.
>
> [1] Boyd, Stephen P., and Lieven Vandenberghe. *Convex optimization*. Cambridge university press, 2004.
>
>
> > In Section 6 related work, the paper discusses offline imitation learning approaches. It claims that these approaches "assume prior knowledge of which data points are generated by experts". However, there are some offline imitation or RL methods which only use the reward signals in the dataset and filter out low-quality data to improve the final performance, such as BAIL [1] and COIL [2]. I am curious about what the advantages of DW are compared to them, especially for BAIL which does not need trajectory information. Also, it will be better if you can try BAIL on your imbalanced dataset and show the results.
> >
> **Answer:**
>
> Thank you for your suggestion! We’ve added BAIL as a baseline and presented the results in Figure 7 in the attached PDF. The results show that DW performs better than BAIL. We are currently running COIL as well and will update the manuscript accordingly when we have the results. Here is the BAIL codebase used in our experiment: https://github.com/lanyavik/BAIL.

---

> > ### Comment · Reviewer_yys4 · 2023-08-14
> >
> > I am very glad to see your new experiment results. It has well proven the effectiveness of your algorithm against other baselines.
> >
> > For the convercency part, I recommend to include or mention this property in the main paper or the appendix for clarity.
> >
> > Now I am also willing to raise my rating. Thanks for your kind reply.

---

> > > ### Author Response · Authors · 2023-08-17
> > >
> > > We thank the reviewer's appreciation of our new results and for raising the rating. We will include these new baselines and analysis on convergence in the updated manuscript.

---

### Official Review · Reviewer_WKvr · 2023-07-06

**Soundness:** 4 excellent
**Presentation:** 3 good
**Contribution:** 3 good
**Rating:** 7
**Confidence:** 4

**Summary:**

The authors consider an offline RL problem where the offline dataset has a small number of high-reward episodes and a larger pool of low-reward episodes. They argue this setting is fairly common in reality, since generating high-reward episodes is often higher effort.

Considering offline RL objective functions that are structured like "maximize reward while staying close to the behavior policy", the argument is that staying close to the behavior policy causes optimization to be too conservative / biased too much towards the large pool of low-reward episodes. Borrowing the DiCE techniques from off-policy evaluation, the proposed method is to learn importance weights to bias the "stay close to data" objective towards the high-reward (expert) trajectories over the support of the original offline dataset. The learning of these weights is done standard to the DiCE style techniques, although in practice the importance weights are given a KL constraint to stay close to the original dataset to avoid collapsing the distribution to a few rare examples, and the importance weights are also applied to the reward-maximizing terms in offline RL (even though theoretically this should not be required).

This is then compared with two other approaches that bias the offline distribution towards higher return episodes - sampling based on the advantage of the offline episode, or sampling the top K% of the offline data. The best results are found by combining advantage weighting with density ratio weighting

**Strengths:**

I have some objections about claiming that offline RL methods have not considered the downsides of constraining to poor return trajectories. (More on this later). But in general, I agree that many datasets often have a smaller number of good examples, importance weights to adjust the conservatism penalty make sense, and using methods from the OPE literature is a reasonable way to learn those importance weights. The evaluation is also extensive enough for me to trust the results.

**Weaknesses:**

There have been methods that attempt to constraint the policy to the support of the offline dataset, rather than the behavior of the offline dataset. BEAR and BRAC come to mind due to using Kernel MMD to measure policy divergence (and they ought to be cited.) In practice these methods have underperformed CQL and TD3+BC so I think it is okay to benchmark against just CQL, but it is still a notable omission.

I also suspect that the KL penalty is one of the more important hyperparams, would need to be tuned separately per dataset (as the acceptable level of deviation will change depending on dataset composition), and there is not much guidance on how to set this parameter. This weakness is common to many offline RL methods though (i.e. the weight of CQL penalty also needs to be tuned).

**Questions:**

Could the authors comment on how the importance weights $w(s,a)$ evolve over time? To me the most dangerous outcome is that $w(s,a)$ learning becomes unstable, or training overfits too heavily due to upsampling a smaller section of the data. Is there any way to compare how the weighting of $w(s,a)$ compares to methods like top K%, in how much they consider different examples in the data?

**Limitations:**

Seems fine.

---

> ### Author Rebuttal · Authors · 2023-08-09
>
> We thank the reviewer’s valuable comments and appreciations on our extensive evaluation. The following address the reviewer’s questions.
>
> > There have been methods that attempt to constraint the policy to the support of the offline dataset, rather than the behavior of the offline dataset. BEAR and BRAC come to mind due to using Kernel MMD to measure policy divergence (and they ought to be cited.) In practice these methods have underperformed CQL and TD3+BC so I think it is okay to benchmark against just CQL, but it is still a notable omission.
> >
>
> **Answer:**
> Thank you for your suggestion! We will cite BEAR and BRAC in the updated manuscript. Meanwhile, we added BEAR as a baseline in Figure 8 in the attached PDF. We ran BEAR based on this public codebase (https://github.com/takuseno/d3rlpy). The results show that both DW-Uniform and DW-AW with CQL and IQL outperform BEAR by a large margin in imbalanced datasets. This indicates that support constraint implemented with kernel MMD may not be able to impose support constraint nicely, hence leading worse performance than CQL with uniform sampling.
>
> > Could the authors comment on how the importance weights $w(s,a)$ evolve over time? To me the most dangerous outcome is that $w(s,a)$ learning becomes unstable, or training overfits too heavily due to upsampling a smaller section of the data. Is there any way to compare how the weighting of $w(s,a)$ compares to methods like top K%, in how much they consider different examples in the data?
> >
>
>
> **Answer:**
> - **Does $w(s,a)$ learning become unstable?** We observed the importance weight $w(s,a)$ converge in the early stage of training (~10,000 gradient steps, which is 1% of training steps). Figure 9 presents the data’s weights at different training epochs, indicating that the weights converge after ten epochs. This means that the weights learned by DW are stable during training rather than oscillating over time.
> - **Does training overfit a small section of the dataset?** Since DW outperforms uniform sampling by a large margin, overfitting may not be a severe issue. Even in the small dataset (Figure 2b) that is prone to have overfitting issues, DW outperforms uniform sampling, which suggests that overfitting is less of a concern.
>
>     **DW can assign finer-grained weights per transition than AW and top K% (i.e., PF in our paper).** We visualize the weights generated under dataset `walker2d-random-medium-small-v2`, where DW outperforms the baselines. See Figure 9 for the plot of the evolution of weight over time and the weights of AW and PF. The color at each index denote the weight of a transition at a particular dataset index (horizontal axis). The left partition (i.e., left of the red line) store the data from medium policy, and the others are data from random policy.
>
>     - **For AW**, we see that only one block of transitions receives weights. It means that all the weights concentrate on the same trajectory. As such, AW is prone to overfit a few trajectories in the dataset.
>     - **For top K%**, it weights each trajectory from medium policy uniformly with the same weight. This may not suffer from overfitting since it’s close to uniform sampling, but it may not improve over uniform sampling a lot.
>     - **Our DW**, in contrast, can assign finer-grained weights per transition, avoiding biasing to only one trajectory and improving the data distribution. This may explain why DW outperforms the baselines in this case. Moreover, we see that the weights of DW converge after epoch 10 (each epoch consists of 1000 gradient steps), which means that the weight evolution is stable.
>
>
> > I also suspect that the KL penalty is one of the more important hyperparams, would need to be tuned separately per dataset (as the acceptable level of deviation will change depending on dataset composition), and there is not much guidance on how to set this parameter. This weakness is common to many offline RL methods though (i.e. the weight of CQL penalty also needs to be tuned).
> >
>
> **Answer:**
> - **Per-dataset KL penalty tuning:** We found that per-dataset tuning is not required. DW outperforms the baselines in imbalanced datasets using the same KL penalty weight across all the datasets. This means that in addition to the hyperparameter of the base offline RL algorithm, we did not observe DW needing any more tuning.
> - **Guidance on setting KL penalty weight $\lambda_K$:**  We suggest using a high KL penalty ($\lambda_K$) for datasets with limited coverage in the state-action space, like expert or medium datasets in D4RL. This prevents undesired weight assignments on state-action pairs that might lead to out-of-distribution next states. If such data get high weights, training can lead to error accumulation and poor policy performance in offline RL [1]. However, setting high $\lambda_K$ does not hinder policy improvement by DW, as datasets with limited coverage also offer fewer opportunities for theoretically-possible maximal policy improvement due to limited rooms for stitching trajectories covering different regions in the state space.
>
>     [1] Liu, Yao, et al. "Provably good batch off-policy reinforcement learning without great exploration." *Advances in neural information processing systems* 33 (2020): 1264-1274.

---

> > ### Comment · Reviewer_WKvr · 2023-08-17
> >
> > Thanks for the rebuttal. I will leave my score unchanged.

---

### Author Rebuttal · Authors · 2023-08-09

We thank all the reviewers’ suggestions and want to highlight **five new baselines** **(Figure 7)** and **two new analysis (Figures 8 and 9)** in the attached PDF file. The following is the summary of our new experiments.

1. We added the comparison to five new prior works, BEAR [1] (Reviewer WKvr), BAIL [2] (Reviewer yys4), SQL and EQL [3], and UWAC [4] (Reviewer XiFQ) in Figure 7 in the attached PDF. The results showed that **our DW-Uniform and DW-AW with CQL and IQL outperform all the newly added baselines on 72 imbalanced datasets.**
2. **Figure 8 (Reviewers** `qcFR` and `XiFQ`)**:** We presented a new analysis to answer why our DW performs better than AW and PF (i.e., top-K% filtering) by showing that DW can stitch trajectories but AW and PF struggle to do so in didactic four-room environment.
3. **Figure 9 (Reviewers `Wkvr`):** we presented the evolution of the distribution of learned importance weight over training epoch, showing that the training of DW is stable.

We list the public baseline implementation used in our experiments in the following:
- BEAR: [github.com/takuseno/d3rlpy](github.com/takuseno/d3rlpy)
- BAIL: [github.com/lanyavik/BAIL](github.com/lanyavik/BAIL)
- SQL/EQL: [github.com/ryanxhr/IVR](github.com/ryanxhr/IVR)
- UWAC: [github.com/apple/ml-uwac](github.com/apple/ml-uwac)

[1] Kumar, Aviral, et al. "Stabilizing off-policy q-learning via bootstrapping error reduction." *Advances in Neural Information Processing Systems* 32 (2019).

[2] Chen, Xinyue, et al. "Bail: Best-action imitation learning for batch deep reinforcement learning." Advances in Neural Information Processing Systems 33 (2020): 18353-18363.

[3] Xu H, Jiang L, Li J, et al. Offline rl with no ood actions: In-sample learning via implicit value regularization[J]. arXiv preprint arXiv:2303.15810, 2023.

[4] Wu Y, Zhai S, Srivastava N, et al. Uncertainty weighted actor-critic for offline reinforcement learning[J]. arXiv preprint arXiv:2105.08140, 2021.

---

### Decision · Program_Chairs · 2023-09-21

**Decision:**

Accept (poster)

**Comment:**

This paper addresses the distributional shift problem in offline RL by optimizing the importance sampling weights, so that the data distribution is similar to that generated by a good policy.  This improves upon existing methods that enforce alignment with the given trajectories, including the low-performing ones.  Experimental results on 77 datasets with varying imbalance demonstrate that the method is more effective than SOTA offline RL algorithms.

This paper is well written, proposing a solid solution to an important problem.  All reviewers, and myself, agree that it is a good addition to the conference. We hope that the reviews are helpful for improving the paper.